# EmbodiedGPT: Vision-Language Pre-Training via Embodied Chain of Thought

**Yao Mu**[1], **Qinglong Zhang**[2], **Mengkang Hu**[1], **Wenhai Wang**[3], **Mingyu Ding**[*,1], **Jun Jin**[4],
**Bin Wang**[4], **Jifeng Dai**[2,5], **Yu Qiao**[2], **Ping Luo**[*,1,2]

[1]The University of Hong Kong, [2]Shanghai AI Laboratory,
[3]The Chinese University of Hong Kong, [4]Noah's Ark Laboratory [5]Tsinghua University

## Abstract

Embodied AI is a crucial frontier in robotics, capable of planning and executing action sequences for robots to accomplish long-horizon tasks in physical environments. In this work, we introduce EmbodiedGPT, an end-to-end multi-modal foundation model for embodied AI, empowering embodied agents with multi-modal understanding and execution capabilities. To achieve this, we have made the following efforts: (i) We craft a large-scale embodied planning dataset, termed EgoCOT. The dataset consists of carefully selected videos from the Ego4D dataset, along with corresponding high-quality language instructions. Specifically, we generate a sequence of sub-goals with the "Chain of Thoughts" mode for effective embodied planning. (ii) We introduce an efficient training approach to EmbodiedGPT for high-quality plan generation, by adapting a 7B large language model (LLM) to the EgoCOT dataset via prefix tuning. (iii) We introduce a paradigm for extracting task-related features from LLM-generated planning queries to form a closed loop between high-level planning and low-level control. Extensive experiments show the effectiveness of EmbodiedGPT on embodied tasks, including embodied planning, embodied control, visual captioning, and visual question answering. Notably, EmbodiedGPT significantly enhances the success rate of the embodied control task by extracting more effective features. It has achieved a remarkable 1.6 times increase in success rate on the Franka Kitchen benchmark and a 1.3 times increase on the Meta-World benchmark, compared to the BLIP-2 baseline fine-tuned with the Ego4D dataset. More demos, code, and dataset information can be found at our [homepage](#).

## 1 Introduction

Embodied AI tasks, e.g., embodied planning, embodied VQA, and embodied control, aim to imbue robots with the ability to perceive, reason, and act within their environment, enabling them to perform long-horizon plans and execute actions autonomously based on real-time observations. Recently, large language models (LLMs) such as GPT4 [1] and PaLM-E [2], have shown promising language understanding, reasoning, and "chain-of-thought" capabilities. Such advances may open new possibilities for developing robots capable of processing natural language instructions, performing multi-modal "chain-of-thought", and planning actions in physical environments.

Large-scale datasets play important roles in training large language models. For example, OpenCLIP trains its ViT-G/14 model on the LAION-2B dataset [3], which contains 2B image-language pairs. Unlike general-purpose visual language tasks that can get a huge amount of weakly labeled image-caption pairs from the Internet, embodied AI tasks require egocentric data in robotics domains. Also,

---

[*]Corresponding authors: Mingyu Ding and Ping Luo ({dingmyu, pluo.lhi}@gmail.com)

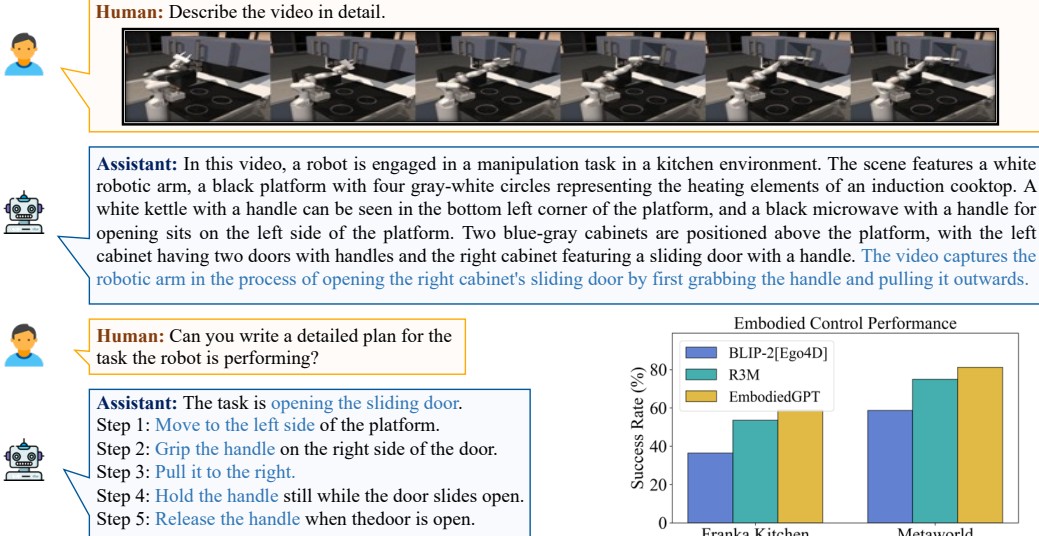

Figure 1: EmbodiedGPT's capabilities for video captioning, multi-turn question answering, embodied planning, and low-level control. The plans given by EmbodiedGPT are highly executable and incorporate task-specific features, leading to a significant improvement in the success rate of embodied control tasks, outperforming both R3M [12] (a video-language contrastive learned model) and BLIP-2 [13] (a multi-modal foundation model) on Franka Kitchen [14] and Meta-World [15] environments.

structured language instructions are needed for precise planning, which usually requires huge manual efforts and costs. This poses a challenging problem in collecting high-quality embodied multi-modal data. Some researchers [4, 5, 6, 7] explore creating large-scale embodied datasets with simulators, but a significant gap remains between simulation and the real world. Recent works [8, 9, 10] also explore adapting the pre-trained LLMs to a new domain by efficient tuning strategies like LoRA [11]. However, several open questions still remain: how to apply LLMs to the field of robotics which may face large domain gaps; how to leverage the "chain-of-thought" capability for structured planning; and how to use the output language plan for downstream manipulation tasks in an end-to-end manner.

To solve the above challenges, in this work, we first build a large-scale embodied planning dataset, termed EgoCOT, which features chain-of-thought planning instructions. It contains carefully selected egocentric videos from the Ego4D dataset [16] and corresponding high-quality step-by-step language instructions, which are machine-generated, then semantics-based filtered, and finally human-verified. Additionally, we also create the EgoVQA dataset as an extension of the Ego4D dataset, focusing on egocentric human-object interaction video question answering tasks, which aims to offer a wider range of egocentric multi-modal data.

Based on our EgoCOT and EgoVQA, we present an end-to-end multi-modal embodied foundation model called EmbodiedGPT, which can interact with the physical world in a more natural and intuitive manner, and perform many embodied tasks, as shown in Figure 1, such as embodied planning, embodied VQA, and embodied control. EmbodiedGPT comprises four integrated modules that work together, including i) a frozen vision model for encoding visual features of current observations, ii) a frozen language model used to execute natural language for question answering, captioning, and embodied planning tasks, iii) an embodied-former with a language mapping layer for aligning the visual and embodied instructions and extracting task-relevant instance-level features with the generated planning for low-level control, and iv) a policy network, which is responsible for producing low-level actions based on the task-relevant features, enabling the agent to effectively interact with the environment. To further enhance EmbodiedGPT 's performance in generating reliable planning containing sub-goal sequences, we implement prefix tuning on the frozen language model to encourage the generation of more executable planning.

Our method possesses the following core advantages: i) the generated planning exhibits strong executability and granularity at the object part level, such as the gripper of a robotic arm or the handle of a door, manifested in sub-goal sequences. ii) the proposed EgoCOT dataset is built based

on an open-source large-scale dataset, which offers greater scalability compared to the PaLM-E [2] model trained on proprietary robot data. And both the EgoCOT dataset, and the EmbodiedGPT model will be open-sourced. iii) EmbodiedGPT forms a closed-loop from high-level planning to low-level control, which enables seamless integration of high-level planning and low-level control, providing efficient task performance and adaptability to a wide range of tasks. To achieve this, we utilize the embodied-former to query task-relevant instance-level features through cross-attention between visual observations and generated embodied planning. This enables the policy network to complete low-level control tasks with fewer than 25 demonstrations.

The contributions can be summarized as follows: (i) We build an end-to-end multi-modal foundation model EmbodiedGPT for embodied AI, which is featured with "chain-of-thought" capability, empowering embodied agents to interact with the physical world in a more natural and intuitive manner. (ii) We develop two datasets, EgoCOT and EgoVQA, consisting of 200M annotated videos from the Ego4D dataset with corresponding detailed planning instructions and VQA data. The datasets are first machine-generated, then semantics-based filtered, and finally human-verified for quality control. (iii) We introduce EmbodiedGPT a cost-effective training approach and a paradigm for extracting task-relevant features from LLM-generated planning queries, thereby forming a closed loop between high-level planning and low-level control. We demonstrate our approach's effectiveness by achieving state-of-the-art or comparable performance on multiple embodied tasks, including embodied control, embodied planning, video captioning, and video QA. Notably, in comparison to BLIP-2 [17] fine-tuned on the Ego4D dataset and R3M [12] specifically designed for manipulation tasks, EmbodiedGPT outperforms both models on the Franka Kitchen [14] benchmark with a margin of 22.1% and 5.5%, respectively. Similarly, on the Meta-World [14] benchmark, EmbodiedGPT surpasses both models with margins of 22.5% and 4.2%, respectively.

## 2  Related Work

### 2.1  Vision Language Pre-training with large scale foundation model

Vision-Language Pre-training focuses on strengthening the link between visual observation and natural language. The goal is to develop models that can better understand and process visual content, such as recognizing objects and actions, and generating descriptive text. As models become larger, the computational expense for end-to-end pre-training rises, leading to the need for modular vision-language pre-training methods. These methods smartly use pre-trained models, keeping them 'frozen' during vision language pre-training to save on computational costs. For example, models like Uniter [18], Oscar [19], VinVL [20], and LiT [21] freeze the image encoder, while Frozen [22] and VGPT [23] freeze the language model. Furthermore, Flamingo [24] and BLIP-2 [17] use both frozen image encoders and language models, providing a balance between performance and computational efficiency. Due to the lack of open-source data for multi-modal embodied planning, previous works struggled to perform detailed task decomposition and lacked the ability to generate precise and executable plans. To tackle this issue, we create the EgoCOT dataset and develop an embodied chain-of-thought vision language pre-training framework to enhance the capacity of multi-modal models for embodied reasoning and planning.

### 2.2  Egocentric Video Datasets.

Egocentric videos, which are captured using wearable cameras, provide a natural perspective of daily activities and pose several challenging research questions [25, 26, 27]. Several egocentric video datasets have been created over the years, including [28, 29, 30]. However, the collection of egocentric videos is expensive, and previous datasets tend to be small-scale and domain-specific. Recently, a massive egocentric video dataset, Ego4D [16], has been released and has been used for embodied representation learning. The dataset comprises 3,670 hours of videos collected by 931 people from 74 locations across 9 countries, with videos accompanied by narrations. For embodied AI tasks, learning from large and diverse egocentric human videos has emerged as a promising approach to acquiring a generally useful visual representation for controlling such tasks. For example, R3M [12] developed a sparse and compact visual representation using the Ego4D human video dataset through a combination of time-contrastive learning and video-language alignment. VIP [31], learns general-purpose reward functions for goal-conditioned robotic manipulation using the Ego4D dataset.

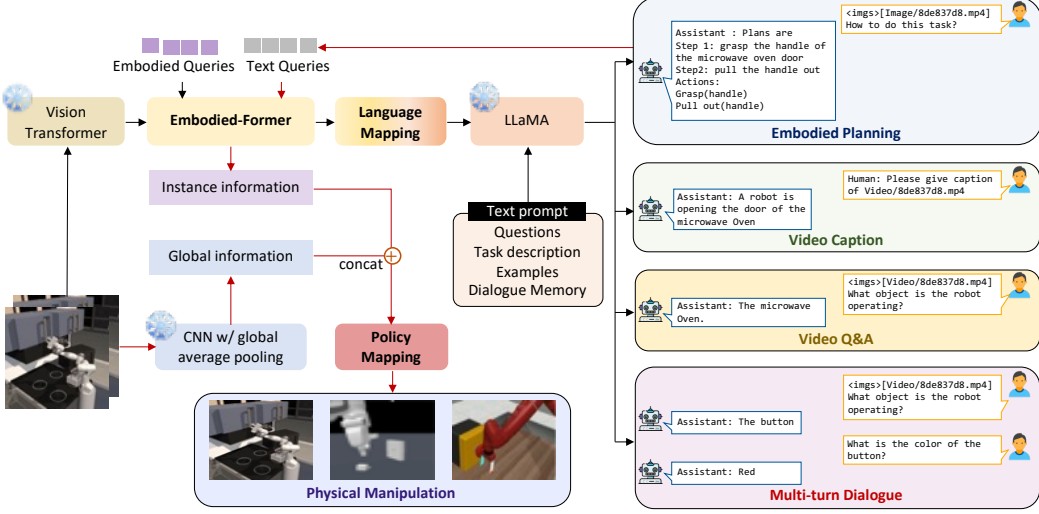

Figure 2: Overall framework of EmbodiedGPT. The black arrow shows the vision-language planning process, while the red arrow represents that we leverage the queried language plans for better policy learning in low-level control tasks.

## 2.3 Large Foundation Model Assistant System

Recent advancements in large-scale multi-modal language models (LLMs), such as GPT-3 [32] and GPT-4 [1], have resulted in the creation of various models that can understand multiple modes of information. Two main approaches are used in this field: systematic collaboration and end-to-end trained models. Systematic collaboration approaches involve coordinating multiple vision models or tools with language models to combine visual information with textual descriptions. Examples include models like Visual ChatGPT [33], MM-REACT [34], and HuggingGPT [35]. However, this approach is limited by the accuracy and capacity of fixed modular models, which can lead to an accumulation of errors. On the other hand, end-to-end models aim to provide unified models for multi-modal tasks. For example, Flamingo [24] combines vision and language by freezing pre-trained vision encoders and language models. BLIP-2 [13] introduces Q-Former to align visual features from frozen visual encoders with large language models. Recently, models such as MiniGPT-4 [36] and LLaVA [37] align instruction-tuned language models with visual features from frozen visual backbones. VideoChat[38], mPLUG-Owl [39] and X-LLM [40], further expand support for video input. PaLM-E [41] is the first large embodied multi-modal model, which directly incorporates features from sensor modalities to improve real-world performance and is trained with their large-scale everyday robot data [42]. Compared to PaLM-E, EmbodiedGPT is more compact, with a size of only 10B and offers additional support for video captioning, video QA and making planning according to a demonstration video. Furthermore, we form a closed-loop system that spans from high-level planning to low-level control.

## 3  Method

The goal of the embodied foundation model is to imitate human-like perception and interaction with the environment by accurately perceiving the environment, identifying relevant objects, analyzing their spatial relationships, and formulating a detailed task plan. To achieve this, the EmbodiedGPT employs a pre-trained vision transformer as the visual encoder and a pre-trained LLaMA [43] model as the language model. As shown in Figure 2, the embodied-former acts as a bridge between the visual and language domains, it first extracts compact visual features from the output of the vision model through attention-based interaction involving visual tokens, text queries, and learnable embodied queries and then maps it to the language modality through a language mapping layer. These embeddings are sent to the frozen LLaMA [43] language model for visual caption, visual QA, and embodied planning. The generated planning is then used to query highly relevant features from the general visual tokens encoded by the visual model via the embodied-former. These features

are utilized to generate low-level control commands for task execution through the downstream policy network. To enhance performance across a range of embodied tasks, we introduce a novel video-language pre-training paradigm that leverages a cognitive chain of thought to produce embodied planning from egocentric video inputs. We formulate this task as a standard VQA (Visual Question Answering) task, using "how to do the task that + original caption" as the question and embodied planning as the answer. This framework enriches the data of embodied planning and standard visual QA tasks, encouraging the embodied-former to capture task-specific features that are more suitable for embodied control tasks.

## 3.1 Framework

The training process consists of three stages, each designed to incrementally develop reasoning and planning capabilities. The first two stages focus on pre-training in basic cognitive and responsive skills, while the third stage involves training the embodied AI task with egocentric video-text data on EgoCOT. In the first stage, we focus on image-text conversation alignment pre-training, which involves using three datasets: COCO Caption [44], 595 thousand finely filtered image-text pairs from CC3M [45], and 491 thousand filtered image-text pairs obtained by re-captioning LAION-400M using BLIP-2 [17]. The primary goal of this stage is to pre-train the Embodied-former and language projection while keeping the vision and language model parameters frozen to save computational resources. In the second stage, our goal is to enhance the model's ability to comprehend and generate more complex sentences and improve its reasoning skills. We achieve this by updating the language projection and prefix language adapter and utilizing the "Complex_Reasoning_77k" and multi-turn conversation datasets provided by "LLaVA_Instruct_150K" [46].

**Embodied "chain-of-thought" training with EgoCOT**: During the third stage, we employ Conv3D [47] to adapt the pre-trained vision model from stage 2 for video encoding, using a total of eight evenly distributed keyframes from each video. Each keyframe is partitioned into three-dimensional (3D) patches, which can be visualized as spatio-temporal cubes, adeptly capturing both the visual content and the sequence of events within the video. These 3D patches are subsequently encoded into visual tokens via the Conv3D module with a time offset of 2 and are then integrated into the internal vision transformer. Then, we introduce the 'chain-of-thought' vision language pre-training paradigm where the model takes 8 keyframes of the video as input, along with the task description, embodied planning, and structured verb-noun pairs summary to reason with a prompt, such as Listing 1. To avoid overfitting, we provide a prompt set that has different instructions with the same meaning. In this stage, we fine-tune the patch embedding, the language projection layer, and the prefix language adapter to better capture temporal information.

```
Watch this video, identify the actions and devise a plan using chain-of-thought. Extract
    detailed actions using this schema:
Task: {"task description"}
Plan: {"plan with chain-of-thought"} Actions: {{"number"}: {'verb'}({'noun'})}.
```

Listing 1: Prompt we used for chain-of-thought pre-training.

## 3.2 Model Architecture

The Embodied-former, denoted as $\mathcal{E}(\cdot)$, serves as a bridge between visual input $x_{\text{vis}}$ and the frozen language model, acting as an information bottleneck that delivers the most relevant visual data to the language model. The Embodied-former consists of two sub-modules: one for extracting features from the image input, denoted as $\mathcal{E}_{\text{vis}} : x_{\text{vis}} \rightarrow y_{\text{vis}}$, and another for extracting features from the text input, denoted as $\mathcal{E}_{\text{txt}} : x_{\text{txt}} \rightarrow y_{\text{txt}}$. We employ $N$ learnable embodied query embeddings $y_{\text{query}}$ as the input of $\mathcal{E}$ to interact with $x_{\text{vis}}$ through cross-attention layers and with $x_{\text{txt}}$ through self-attention layers. We denote the output query representation as $z \in \mathbb{R}^{N \times D}$, where $D$ is the dimensionality of the embeddings. The dimension of $z$ is significantly smaller than that of the visual features. The output query embeddings are then transformed to $z^{'} \in \mathbb{R}^{N \times D^{'}}$, which have the same dimensionality $D^{'}$ as the LLM's text embedding in the language modality. This transformation is performed by a mapping function denoted as $M : z \rightarrow z^{'}$, which is accomplished by a linear projection via a fully-connected (FC) layer. The projected embeddings, $z'$, serve as "soft visual prompts for the language model," decoupling the whole interaction into visual-query interaction and

query-text interaction. The final embodied planning is inferred by the language model with $z'$ and text prompt(shown as Listing 1) as input. For low-level control which aims to generate actions to interact with the environment, the embodied plan $x_{\text{plan}}$ is used as input text for embodied-former to query the task-relevant instance level features $z_{\text{instance}} = \mathcal{E}(x_{\text{vis}}, x_{\text{plan}}, y_{\text{query}})$. Subsequently, the agent is capable of generating control commands, such as the turning angle of the servo, represented as $a = g(z_{\text{instance}}, z_{\text{global}})$. This function combines both the instance-specific information $z_{\text{instance}}$ and the global context $z_{\text{global}}$. The global context is inferred using a ResNet50 model [48] that has been pre-trained on ImageNet [49], employing global average pooling. Here, $g(\cdot)$ represents the policy network, which is a Multi-Layer Perceptron (MLP) [50] mapping function. The output of the policy network consists of specific executable actions, such as positions and velocities in the Cartesian coordinate system. More implementation details can be found in Appendix A.

### 3.3 Training Settings

We employ the same pre-trained image encoder as BLIP-2[17]. Specifically, we utilize the ViT-G/14 model from EVA-CLIP [51] and remove its last layer, using the output features of the second last layer instead. For the frozen language model, we adopt a pre-trained LLaMA-7B [43] model and fine-tune it using the ShareGPT dataset and a GPT-4 generated 52K English instruction-following dataset [52]. We then utilize the well-fine-tuned language model as the frozen language model for vision-language pre-training. Additionally, we convert the data type of parameters of the frozen ViT [53] and language model to FP16 during pre-training to increase efficiency.

### 3.4 Creating EgoCOT and EgoVQA Dataset

For our EgoCOT dataset, we obtain basic data from the Ego4D dataset [16], which includes $9,645$ untrimmed videos of various durations ranging from 5 seconds to 7 hours. To prepare the data for our purposes, we conducted two stages of data cleaning to prepare our data. In the first stage, we filtered out videos with missing or very short narrations (which made up 7.4% and 0.9% of the text, respectively), as well as those with unsure tags (which accounted for 4.0% of the text). We also excluded videos without human-object interaction, such as watching TV or walking. After this stage, we were left with 2.9 thousand hours of video, containing 3.85 million narrations, from 129 different scenarios covering 2927 hours of video.

To generate pairs of captions, embodied plannings, and corresponding video segments with time intervals, we utilized the EgoVLP framework [54] to segment the video. The narrations are organized as a sequence of sentences $\mathcal{T}_0, \cdots, \mathcal{T}_n$ with precise timestamps $t_0, \cdots, t_n$ that indicate when a described event occurred. For each narration $\mathcal{T}_i$ with timestamp $t_i$, we paired it with a clip $\mathcal{V}_i$ by determining its start and end time points:

$$[t_i^{start}, t_i^{end}] = [t_i - \beta_i/2\alpha,\ t_i + \beta_i/2\alpha], \tag{1}$$

where $\beta_i = \sum_{j=0}^{n-1} (t_{j+1} - t_j)/n$ is an adjustable parameter equal to the average temporal distance between consecutive narrations in a given video. Conversely, $\alpha$ is a scale factor computed as the average of all $\beta_i$ across all videos in the EgoCOT dataset ($\alpha = 4.9$ seconds). For each video segment, we provide prompts and corresponding captions for ChatGPT [55] to generate a reasonable and detailed embodied planning. The caption is typically a brief introduction such as "C opens a drawer." We use the ChatGPT to generate a chain of thought according to the caption and organize it into a list of verb-noun pairs, such as *"plans: grasp the handle with the gripper and pull the handle; actions: 1. grasp(handle, gripper) 2. pull(handle)."* The prompt we used to generate EgoCOT dataset is shown in Listing 2. To enhance the diversity of generated chain of thoughts, we employ a temperature parameter of 0.9 and a top-p parameter of 0.95. For each prompt, we perform five sampling iterations.

**Post-procedure.** To ensure the quality of the generated planning instructions, we perform the second stage of data cleaning. We used the CLIP model [56] to assess the similarities between the video and text pairs. For each video, we compared it against five potential embodied plans and selected the one with the highest similarity as the corresponding label for the embodied plan. We then took our data-cleaning process a step further by filtering out any video-caption-planning pairs with similarities lower than the threshold. We eliminated both data with the low similarity between the video and caption and between the video and planning to ensure the highest quality data for our EgoCOT dataset. For each keyframe of the video segment, we use the CLIP model to encode both the text data $T$ and the image data $I$ into a shared embedding space. The similarity is calculated using the

```
You need to generate plans with chain of thought for each task, and then extract
    detailed actions (collocation of nouns and verbs) from the plan.
The action can be of the following form:
[action_name], eg., turn left;
[action_name] argument1, eg., pick up(apple);
[action_name] argument1 argument2, eg., put(apple, table)
Task: pick up a cup on the table
plans: grasp the handle of the cup with the gripper and lift it up
Actions:
1. grasp(handle of the cup, gripper)
2. lift up(cup)
```

Listing 2: Prompt we used for creating EgoCOT dataset.

cosine similarity function as $S(y_T, y_I) = \frac{y_T \cdot y_I}{\|y_T\|\|y_I\|}$, where $S(y_T, y_I)$ denotes the similarity between the text and image, and $y_T$ and $y_I$ are the respective embeddings. Given that each video contains multiple keyframes, an ensemble of similarity scores is obtained for each video. This ensemble strategy helps to alleviate the problem of variability among individual frames and ensures a more robust and representative measure of overall similarity. The ensemble similarity score between a video $V$ with $n$ keyframes and text data $T$ is given by:

$$E(V, T) = \frac{1}{n} \sum_{i=1}^{n} S(y_{Ti}, y_{Ii}) \tag{2}$$

where $E(V, T)$ is the ensemble similarity score, $S(y_{Ti}, y_{Ii})$ is the similarity score for the $i$-th keyframe, and $n$ is the total number of keyframes. We also created the EgoVQA dataset specifically for egocentric human-object interaction video question answering tasks to enrich the training data. For each caption in the Ego4D dataset, we used ChatGPT to generate five QA pairs. To ensure relevance, we guided ChatGPT to focus on core key verbs and nouns by designing prompts as shown in Listing 3. The sampling schema when crafting EgoVQA is the same to that as EgoCOT.

```
Please ask some questions accroding to the verbs and nouns in the sentence.
For example, in this sentence "a man is picking up a cup", the verb is picking up and the
    noun is cup, therefor questions can be "what is the object the man is picking up?"
    or "what operation is performed on the cup?".
Then You need to give the answer.

input: a man is picking up a cup
question: What is the object the man is picking up
answer: The cup
```

Listing 3: Prompt used for creating EgoVQA dataset.

# 4 Experiments

In this section, we present a comprehensive evaluation of multi-modal foundation models and EmbodiedGPT, across various tasks including visual captioning, embodied planning, and control.

**Evaluation on image input tasks.** In order to evaluate the quality of generated captions and planning with the given image, we conducted a user study with 30 participants. The study included 10 cases of image caption tasks from MS-COCO dataset [44], 5 embodied planning scenarios in different embodied AI simulators, and 5 real-world scenes with accompanying embodied planning tasks. Participants were asked to rate the generated captions from different end-to-end models on five dimensions using a scoring system ranging from 1 to 10: object recognition accuracy, spatial

| Model | Object(↑) | Spatial(↑) | Redundancy(↓) | Plan Reasonable(↑) | Plan Executable(↑) |
|-------|-----------|------------|---------------|--------------------|--------------------|
| Minigpt4 | 5.6 | 4.8 | 4.4 | 4.5 | 4.8 |
| LLaVA-7B | 7.3 | 7.4 | 3.9 | 7.5 | 6.6 |
| LLaVA-13B | **8.5** | 8.6 | 3.4 | 8.4 | 7.6 |
| EmbodiedGPT | 8.4 | **8.8** | **2.6** | **8.8** | **8.4** |

Table 1: Generate Quality Evaluation on image input tasks.

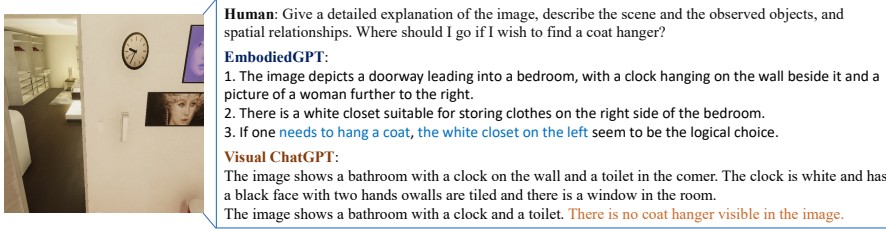

**Human**: Give a detailed explanation of the image, describe the scene and the observed objects, and spatial relationships. Where should I go if I wish to find a coat hanger?

**EmbodiedGPT**:
1. The image depicts a doorway leading into a bedroom, with a clock hanging on the wall beside it and a picture of a woman further to the right.
2. There is a white closet suitable for storing clothes on the right side of the bedroom.
3. If one needs to hang a coat, the white closet on the left seem to be the logical choice.

**Visual ChatGPT**:
The image shows a bathroom with a clock on the wall and a toilet in the corner. The clock is white and has a black face with two hands owalls are tiled and there is a window in the room.
The image shows a bathroom with a clock and a toilet. There is no coat hanger visible in the image.

Figure 3: Comparison between EmbodiedGPT and VisualGPT in the question-answering task.

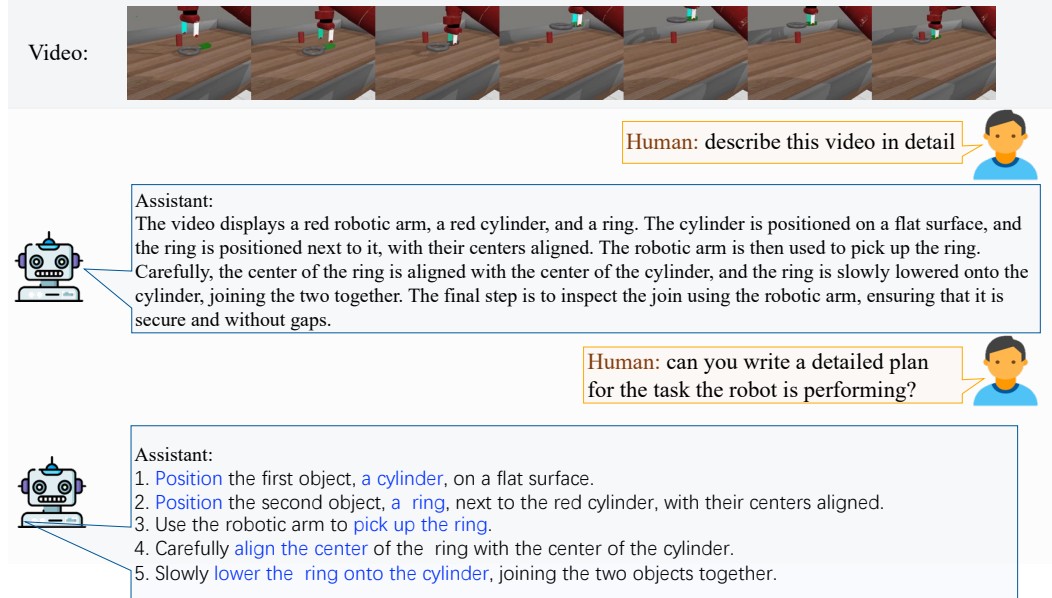

Figure 4: Example of video input embodied AI tasks on Meta-World benchmark. EmbodiedGPT accurately analyzes embodied control tasks in demonstration videos and provides precise planning.

relationship understanding, level of redundancy in the answer, and reasonability of the planning and the executability of the planning. The average scores among all the participants for different models are shown in Table 1. The results demonstrate that EmbodiedGPT achieves a comparable level of object recognition and spatial relationship understanding as the LLaVA-13B model, despite having only 7B parameters in the language model. Furthermore, EmbodiedGPT generates less redundant content in relation to the given embodied AI task, and produces the most reasonable and executable planning outputs. We also compared the performance of EmbodiedGPT with Visual ChatGPT [33], which adopts a hierarchical approach by combining several pre-trained vision models and language models to answer questions. In the Virtual-Home [57] benchmark, Visual ChatGPT uses a visual caption model to generate dense captions that are subsequently passed into ChatGPT for deriving a solution. As shown in Figure 3, Visual ChatGPT failed to find a coat hanger due to its limitations of relying solely on the caption model for extracting visual information, resulting in poor performance when compared to the end-to-end model like EmbodiedGPT. These findings highlight the advantages of adopting a unified, end-to-end model over hierarchical approaches that rely on multiple stages.

**Evaluation on video input embodied AI tasks.** We evaluate the recognition ability of videos and planning abilities of our model for embodied control tasks on standard embodied AI benchmarks, Franka Kitchen [14] and Meta-World [15]. Meta-World provides a challenging set of tasks that require complex object manipulation skills, including assembling a ring on a peg, picking and placing a block between bins, pushing a button, opening a drawer, and hammering a nail. Franka Kitchen benchmark focuses on tasks like sliding open the right door, opening the cabinet, turning on the light, turning the stovetop knob, and opening the microwave. As shown in Figure 4, given a demonstration video, EmbodiedGPT can accurately interpret the embodied control task and provide step-by-step planning. The output planning is fed into the Embodied-former module of EmbodiedGPT to query

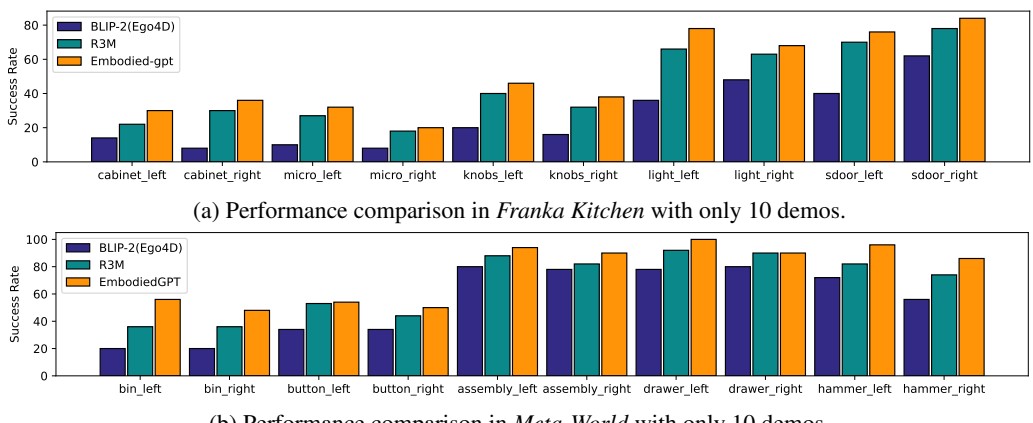

(a) Performance comparison in *Franka Kitchen* with only 10 demos.

(b) Performance comparison in *Meta-World* with only 10 demos.

Figure 5: Performance of EmbodiedGPT in low-level control tasks with 10 demonstration demos.

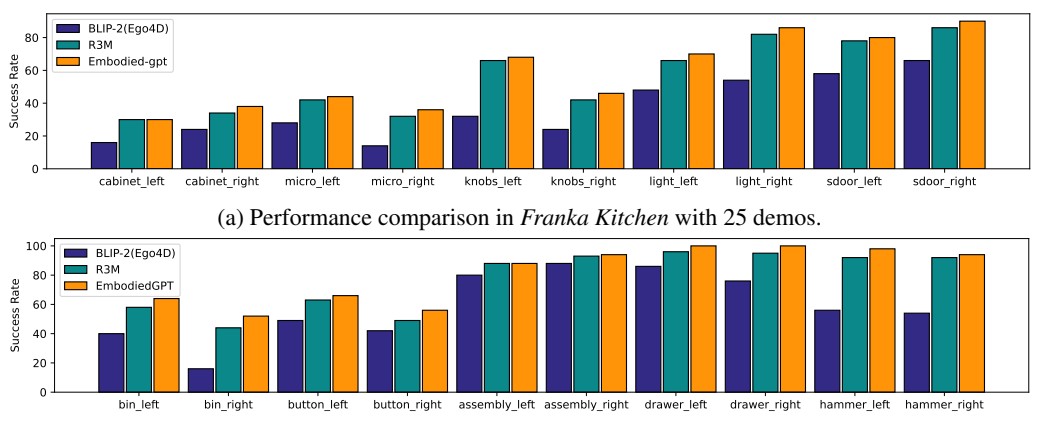

(a) Performance comparison in *Franka Kitchen* with 25 demos.

(b) Performance comparison in *Meta-World* with 25 demos.

Figure 6: Performance of EmbodiedGPT in low-level control tasks with 25 demonstration demos.

highly relevant features for use as inputs in the policy network and the low-level actions are generated by the policy network to interact with the environment (see more visualizations in Appendix B).

**Evaluation on embodied control tasks.** For embodied control tasks, we compare our model with R3M[12], which is the state-of-the-art method in these two benchmarks, and an ablation version called 'BLIP-2[Ego4D]', which has the same structure and same amount of parameters as EmbodiedGPT, and is only fine-tuned on the video caption task using the Ego4D dataset without incorporating EgoCOT. In all experiments, the policy network is learned using few-shot learning on a small amount of demonstration data. There are two settings, one of which utilizes 10 demonstrations, and the other utilizes 25 demonstrations. We report the success rate in 100 random evaluations with only visual observations in 5 tasks per benchmark over 5 seeds and 2 different camera views for each setting, respectively. As shown in Figure 5 and Figure 6, EmbodiedGPT outperforms the baseline methods, demonstrating the effectiveness of learning with EgoCOT.

| Model | Franka(10 demos) | Franka(25 demos) | Meta-World(10 demos) | Meta-World(25 demos) |
|---|---|---|---|---|
| EmbodiedGPT | **50.8**% ±2.8 | **58.5**% ±2.7 | **76.4**% ±2.2 | **81.2**%±2.0 |
| - Close-loop | 38.6% ±2.9 | 47.3% ±2.5 | 62.7% ±2.2 | 64.9% ±2.0 |
| - COT | 26.2% ±3.2 | 36.4% ±2.7 | 55.2% ±2.4 | 58.7% ±2.0 |

Table 2: Ablation on the closed-loop spans from planning to low-level control, and "chain-of-thought" (COT) training with 25 and 10 demonstrations("-" symbol indicates "removing"). We report the average success rate over 5 tasks and 2 camera views per benchmark.

**Human**: You are the controller of the robot shown in this image, now your task is put all the objects into the brown bag next to the robot, please write a step by step plan:

**EmbodiedGPT**:
Step1. Move the robotic arm to the top of the gray bottle.
Step 2. Close the gripper and grasp the gray bottle.
Step 3. Move the arm to the brown bag.
Step 4. Release the bottle.
Step 5. Move the arm to the top of the cereal bag.
Step 6. Close the gripper and grasp the cereal bag.
Step 7. Move the arm to the brown bag.
Step 8. Finally, release the cereal bag.

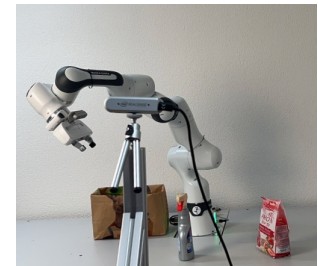

Franka Emika Panda arm with front facing camera

Figure 7: Generated plan by EmbodiedGPT for rearrangement task.

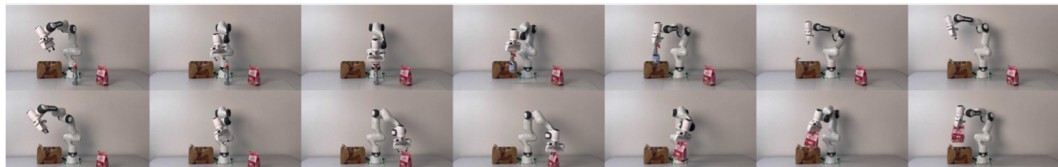

Figure 8: Real-world experiment on the Franka Emika robot arm.

**Ablation study.** We perform ablation studies to analyze the effectiveness of the "Chain-of-Thought" training mode and the importance of a closed-loop design for embodied control. The results, as shown in Table 2, demonstrate a significant improvement in success rate when using the EgoCOT approach compared to training solely with the EGO4D caption task. Moreover, the closed-loop design is necessary as the generated plans contained specific and relevant sub-goal information, which proved crucial for control tasks. In summary, EmbodiedGPT exhibits a strong ability to generate reasonable planning, accurately extract task-relevant features from visual inputs, as well as execute low-level actions to interact with the environment. The ablation experiments demonstrate that both the training paradigm based on EgoCOT and the closed-loop design from embodied planning to low-level control significantly contribute to the performance improvement of EmbodiedGPT.

**Real robot experiment.** We also conducted a real-world experiment using the Franka Emika robot arm, emphasizing a rearrangement task where the objective was to pack scattered bottles from a table into a box. As depicted in Figures 7 and 8, *model* is capable of generating intricate plans and executing of low-level actions based on 50 demonstration.

## 5   Conclusion

In this paper, we present EmbodiedGPT, an end-to-end multi-modal foundational model for embodied AI that enables agents to perform step-by-step planning and execute low-level commands. To achieve this, we create a large-scale embodied planning dataset called EgoCOT and develop an efficient training approach that utilizes prefix tuning to generate high-quality plans with a "chain-of-thought". Furthermore, our embodied control paradigm seamlessly coordinates high-level planning and low-level control. Extensive experiments demonstrate the effectiveness of EmbodiedGPT on various embodied tasks, achieving state-of-the-art or comparable performance. We believe that EmbodiedGPT represents a significant step towards developing more intelligent embodied AI agents.

**Future works and limitations:** EmbodiedGPT freezes the parameters of the vision and language model due to limited computational resources. Joint training with all modules and exploring other modalities, such as speech, could be future works. We do not foresee obvious undesirable ethical or social impacts at this moment.

**Acknowledgement.** This paper is partially supported by the National Key R&D Program of China No.2022ZD0161000 and the General Research Fund of Hong Kong No.17200622. We would like to express our appreciation to Dr. René Zurbrügg from ETH AI Center for his support on robot experimental platform.

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
