# Appendix for EmbodiedGPT: Vision-Language Pre-Training via Embodied Chain of Thought

**Yao Mu**[1], **Qinglong Zhang**[2], **Mengkang Hu**[1], **Wenhai Wang**[3], **Mingyu Ding**[*,1], **Jun Jin**[4],
**Bin Wang**[4], **Jifeng Dai**[2,5], **Yu Qiao**[2], **Ping Luo**[*,1,2]
[1]The University of Hong Kong, [2]Shanghai AI Laboratory,
[3]The Chinese University of Hong Kong, [4]Noah's Ark Laboratory [5]Tsinghua University

## A Implementation details

### A.1 Hyper-parameters

We use the same set of training hyper-parameters for all models during vision-language pre-training. We employ the AdamW optimizer [1] with $\beta_1 = 0.9$, $\beta_2 = 0.98$, and a weight decay of 0.05. We utilize a cosine learning rate decay with a peak learning rate of $2 \times 10^{-5}$ and a linear warm-up with a warm-up ratio of $5 \times 10^{-2}$. Our training data consists of images of size 224×224 that are augmented with random resized cropping and horizontal flipping. The maximum sequence length of the language model is set as 256.

### A.2 Video encoding

Firstly, videos are distilled into distinct keyframes. These keyframes are then partitioned into three-dimensional (3D) patches. Leveraging the Conv3D technique, these 3D patches undergo a transformation into visual tokens. In the concluding step, these visual tokens seamlessly integrate with the vision transformer's core architecture.

Specifically, we extract eight evenly distributed keyframes from each video. These keyframes are utilized as the primary inputs for our Conv3D module. The video gets divided into multiple three-dimensional (3D) patches, where each patch has a size defined by $patch\_size \times patch\_size \times frame\_stride$. As shown in Listing 1, these 3D patches are then embedded into visual tokens through the Conv3D module. Each of these 3D patches signifies a spatio-temporal cube within the video, thus efficiently capturing the visual content and the sequence of scenes, and incorporating the temporal dynamics of the videos.

```
patch_embedding = nn.Conv3d(
    in_channels=3,
    out_channels=embed_dim,
    kernel_size=(frame_stride, patch_size, patch_size),
    stride=(frame_stride, patch_size, patch_size))
```

Listing 1: Pseudocode of the Conv3D for video encoding

In this code, the `in_channels` parameter is set to 3, representing the Red, Green, and Blue (RGB) channels of the video. The *out_channels* parameter corresponds to the dimensions of the embedding. The *kernel_size* and *stride* parameters specify the size and stride of the 3D patch, respectively.

After the division of the video into 3D patches, Conv3D captures the spatio-temporal features from the frame sequence and converts each keyframe into a set of visual tokens. These visual tokens then serve as inputs for the vision transformer encoder. The vision transformer leverages the temporal

---

[*]Corresponding authors: Mingyu Ding and Ping Luo ({dingmyu, pluo.lhi}@gmail.com)

37th Conference on Neural Information Processing Systems (NeurIPS 2023).

information within the visual tokens derived from the keyframes, which allows for effective encoding of the video content. By processing this sequence of visual tokens, we adapt the pretrained image encoder to manage video inputs efficiently.

### A.3  Downstream policy learning

We adopt imitation learning as the method of policy learning in low-level control tasks, which leverages demonstration data provided by an expert to learn the desired behavior. This technique has found applications in various domains, such as robotics, autonomous driving, and game playing. We provide each task with 25 demonstrations, which are trajectories of observations and actions performed by an expert in the given task, and test the performance with 25 demonstrations and only 10 demonstrations respectively. The goal of imitation learning is to learn a policy, denoted as $\pi$, that maps the agent's observations to appropriate actions. The learned policy should be able to imitate the expert's behavior accurately. Specifically, we use behavioral cloning to learn the downstream policy, which trains a supervised learning model to predict actions given states based on the expert demonstrations, and the loss function is shown as in Equation 1:

$$L(\theta) = \sum [\pi_\theta(a|s) \log P^*(a|s)] \tag{1}$$

Here, $\theta$ represents the parameters of the policy model, $\pi_\theta(a|s)$ denotes the predicted action probability distribution given a state $s$, and $P^*(a|s)$ represents the ground truth action probability distribution derived from the expert demonstrations.

For Franka Kitchen [2] tasks, the length of a demonstration is 50, which contains 50 state-action pairs. For Meta-World [3] tasks, the length of a demonstration is 500, which contains 500 state-action pairs. Our evaluation methodology is loosely inspired by R3M [4], but we only use visual observations to assess the effectiveness of EmbodiedGPT. We employ a visual representation $z_t$ generated by EmbodiedGPT and train the policy $\pi$ using a standard behavior cloning loss $||a_t - \pi(z_t)||_2^2$. The parameterization of $\pi$ consists of a two-layer MLP [5] with a preceding BatchNorm [6] at the input. The agent is trained for 20,000 steps, and we evaluate its performance in the environment every 1000 steps, reporting the best success rate achieved. For each task, we conduct 5 behavior cloning runs with different seeds. The final success rate reported for a method on a specific task represents the average across 5 seeds, 2 camera viewpoints, and 2 demo dataset sizes, resulting in a total of 20 runs.

## B  More demos of EmbodiedGPT

### B.1  Visual Captioning

We assessed EmbodiedGPT on numerous visual captioning tasks spanning a range of embodied AI benchmarks. As shown in Figure 1, the model displayed an exceptional ability to accurately describe objects, characters, and spatial relationships relevant to embodied AI tasks. Furthermore, EmbodiedGPT exhibited robust zero-shot learning capabilities, evidenced by its strong performance across multiple benchmarks without the need for task-specific fine-tuning.

### B.2  Embodied Planning with video input

Here, we present additional examples of embodied planning with video inputs. As shown in Figure 2, the examples demonstrate that EmbodiedGPT can generate high-quality and executable planning.

### B.3  Embodied Planning with image input

**Embodied Planning for Concrete Tasks**: In the context of concrete task planning, such as making a cup of coffee, EmbodiedGPT effectively utilized visual information to pinpoint the required objects and their positional relationships within the environment. As shown in Figure 3, EmbodiedGPT produced coherent, multi-step plans, taking into consideration both the task requirements and environmental constraints. This capability demonstrates the model's potential to facilitate real-world applications and tackle complex planning challenges within the realm of embodied AI.

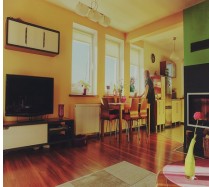

Human: Give a detailed explanation of the image, describe the scene and the objects, and spatial relationships

Assistant:
The image depicts a cozy living room with a woman standing in the middle of it. A large dining table is located in the room, surrounded by chairs. A television is mounted on the wall, and there are multiple potted plants on the windowsill in the dining area. A vase is also present on the table, adding a touch of decoration to the room. The living room appears to be well-furnished and comfortable for relaxation and socializing. The kitchen area and the living area appear to be connected, with the dining table located in the middle and the kitchen counter adjacent to it.

(a) Image caption in COCO dataset.

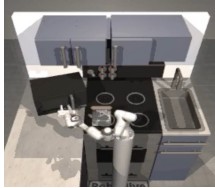

Human: Give a detailed explanation of the image, describe the scene and the objects, and spatial relationships

Assistant:
1. The image depicts a small kitchen area where a robot is cooking near the kitchen counter.
2. In the middle of the kitchen area, there are several appliances and objects. A black stove, a white water kettle, and a microwave oven are located on the left side.
3. The robot is actively engaged in the cooking process and can be seen near the stove.
4. The kitchen backsplash is made of white tiles, providing a clean and neat appearance. Along the walls, there are cabinets for storage purposes.
5. On the right side of the image, there is a sink, serving as a water source for the kitchen activities.
6. The kitchen counter, where the robot is cooking, is positioned on the left side of the image.

(b) Embodied image caption in Franka Kitchen benchmark.

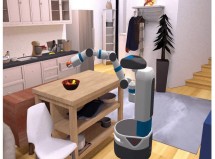

Human: Give a detailed explanation of the image, describe the scene and the objects, and spatial relationships

Assistant:
1.  The image depicts a clean and well-equipped kitchen with various appliances and furniture.
2.  A robot is present, standing next to the kitchen counter and interacting with a bowl on the countertop.
3.  The robot's arm is positioned over the table, and its hand is holding a spoon filled with food. The robot is likely a kitchen helper, possibly designed to assist with meal preparation.
4.  The kitchen counter itself is well-stocked, featuring a sink, a refrigerator, and a dining table. The dining table and a chair are positioned in the center of the room, providing a convenient space for people to gather and enjoy their meals.

(c) Embodied image caption in Habitat2.0 benchmark.

Figure 1: Generation results in image caption tasks.

**Embodied Planning for Abstract Tasks**: For abstract task scenarios, EmbodiedGPT adeptly combined visual observations with abstract concepts to generate concrete sub-task descriptions. For instance, as shown in Figure 4, when given the abstract prompt of feeling hot, the model identified pertinent objects in the environment (e.g., a fan) and suggested a practical solution (e.g., turning on the fan). Subsequently, the model generated a detailed plan to accomplish the identified sub-tasks, highlighting its adaptability across diverse problem-solving contexts.

## C   Evaluation metric and scoring criteria for user study

We show Table 1 to outline the scoring criteria for a user study incorporating the above five evaluation metrics:

## D   Insight about the prompt designing for multi-modal large model

Through extensive experiments evaluating multi-modal large models in question answering tasks, we have identified a limitation in their ability to effectively utilize visual information. These models tend to rely heavily on textual inputs and may not fully exploit the visual cues present in the question-answering process. In order to address this limitation and enhance their performance, we aimed to investigate the impact of incorporating additional prompts that guide the model's attention toward the visual content.

**Human:** Describe the video in detail.

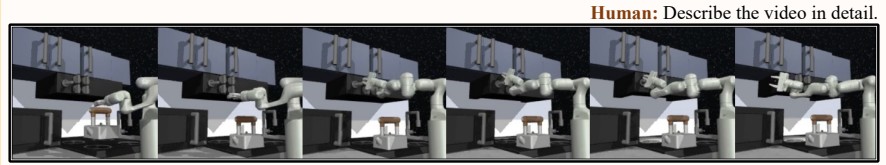

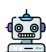

**Assistant:** This video depicts a robot performing a manipulation task in a kitchen backdrop. The scene includes a white robotic arm, a black platform with four gray-white circles that indicate the heating zones of an induction cooktop. In the lower left corner of the platform, there is a white kettle with a handle, and on the far left, a black microwave oven with a handle for opening. Above the platform, two blue-gray cabinets are situated; the left cabinet has two doors with handles, and the right cabinet has a sliding door with a handle. Below the cabinets, there are several knobs and a rod-shaped switch that can be toggled. The switch is located on the far left, while the other knobs are on the right. In the video, the robotic arm is moving the rod-shaped switch to the left to illuminate the workspace.

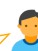

**Human:** What task is the robot doing?

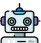

**Assistant:**
The robotic arm is moving the rod-shaped switch to the left to illuminate the workspace.

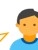

**Human:** Can you write a detailed plan for the task the robot is performing?

**Assistant:**
Step 1: Identify the rod-shaped switch on the platform
Step 2: Approach the switch and ensure that it is not too hot to touch.
Step 3: Grasp the switch and pull it to the left, moving it to the on position.
Step 4: This action will turn on the lights in the kitchen, providing a brighter workspace.

(a) Embodied planning for the light opening task in Franka Kitchen.

**Human:** Describe the video in detail.

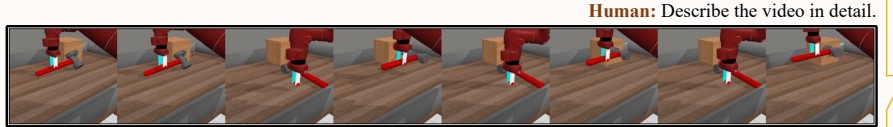

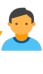

**Assistant:** The video features a red robot with a mechanical arm, a yellow table, a hammer with a red handle, and a yellow wooden box with a nail protruding from it. The robot is hammering the nail into the box. The mechanical arm first picks up the hammer, aligns the gray iron head with the nail, hammers it, and then slowly lowers the hammer.

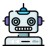

**Human:** Can you write a detailed plan for the task the robot is performing?

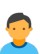

**Assistant:**
Step 1: Identify wooden box and nail.
Step 2: Move close to hammer.
Step 3: Grasp hammer and lift until firm.
Step 4: Align hammer head with nail.
Step 5: Strike nail with hammer.
Step6: Slowly lower hammer.

(b) Embodied planning for hammering nail task in Meta-World.

Figure 2: Embodied planning with video input.

Human: Give a plan to make a coffee based on the image and explain it in detail.

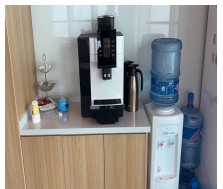

Assistant:
1. First, locate the coffee maker, which is a white coffee machine sitting on a wooden counter near the water dispenser.
2. Next, take a closer look at the coffee maker to find the buttons and functions, such as the power button, water reservoir, and the selection button.
3. Ensure the coffee maker is turned on.
4. Fill the water reservoir with water, either by using a tap or a bottle, and select the type and amount of coffee to brew. Depending on the coffee maker, you can select the number of cups, the strength, or the type of coffee to brew.
5. After selecting the desired coffee, press the \"brew\" button. The coffee maker will then brew the selected coffee, which will fill up a cup positioned on a nearby counter.
6. Pour the coffee into the desired cup using the available pouring spout of the coffee maker.
7. Once the coffee has been poured, you can add any desired ingredients, such as sugar or milk, to enhance the taste.
8. Finally, enjoy your coffee while taking a moment to appreciate the convenience and functionality of the coffee maker."

Figure 3: Embodied planning for concrete tasks with image input.

Human: With the objects in the image, if you want to meeting with someone, where will you go? Write your plan and explain it.

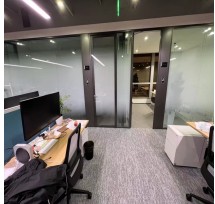

Assistant:
1. Identify your meeting point. In this image, there are two glass sliding doors with a large window visible between them. One can be the meeting point, and the other can be the entrance or exit of the room.
2. Choose a meeting room. Since there is a room with a desk, chair, and computer, you can use this as a meeting room.
3. Make a meeting schedule. Since there is no one in the room, schedule your meeting at a convenient time for both you and the person you'll be meeting with.
4. Prepare your materials. Since there is a desk and chair in the room, you can use them to prepare your materials for the meeting.
5. Arrange your meeting space. To make the meeting more professional, you can rearrange the furniture in the room to face the large window. This will provide a natural light and a pleasant atmosphere.
6. Invite your guests. After you have selected your meeting point and organized your materials, send an invitation to your guests to let them know about the meeting time, location, and any other relevant information."

(a) Embodied planning for abstract tasks [meeting].

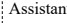

Human: With the objects in the image, if you feel hot, what will you do, write your plan and explain it.

Assistant:
1. First, find the object that is likely to be cool, which in this case is a fan.
2. Then, position the fan facing towards yourself to blow cool air towards you.
3. Finally, adjust the fan's settings to increase the airflow and cooling effect, if needed.
4. By following these steps, you can use the items in the image to make yourself cool, maintain comfort, and minimize the discomfort caused by high temperatures.
5. By following these steps, you can use the items in the image to make yourself cool, maintain comfort.

(b) Embodied planning for abstract tasks[feel hot].

Figure 4: Embodied planning for abstract tasks with image input.

To achieve this goal, we propose a straightforward yet highly effective approach: incorporating additional prompting into the model's input. This involves introducing specific prompts such as "in the scene shown in this image/video" or allowing the model to describe the image/video as part of a multi-turn dialogue. By including these prompts, we aim to explicitly direct the model's focus toward the visual information available and encourage it to utilize this information when generating answers. Our experiments have yielded promising results. As shown in Figure 5, the introduction of additional prompts has significantly improved the model's ability to leverage visual information and provide accurate answers based on the visual content. By explicitly referencing the scene depicted in the image or video, the model's attention is directed toward the relevant visual features, leading to a more comprehensive integration of visual and textual information. Consequently, the model's reasoning ability is enhanced, resulting in more precise and contextually grounded answers.

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

| Evaluation Metric | Explanation |
|---|---|
| Object Recognition Accuracy | This metric measures the ability of a system to accurately identify objects from images or videos. A higher accuracy indicates that the system can correctly recognize the objects present in the given visual data. |
| Spatial Relationship Understanding | Spatial relationship understanding refers to the system's capability to accurately discern the spatial relationships between objects in a scene. It evaluates whether the system can determine the relative positions, orientations, distances, and other spatial attributes of objects with precision. |
| Level of Redundancy in the Answer | The level of redundancy in the answer assesses the amount of unnecessary or repetitive information present in the system's response. Lower redundancy indicates that the system provides concise and non-repetitive answers, which is generally preferred as it reduces verbosity and improves clarity. |
| Reasonability of the Planning | The reasonability of the planning metric gauges the logical coherence and appropriateness of the system's planning process. It examines whether the system's generated plans are sensible and align with the given goals or objectives. |
| Executability of the Planning | This metric evaluates the feasibility and practicality of the system's generated plans. It assesses whether the plans can be executed successfully in the real world. |

Table 1: Evaluation metric and scoring criteria for user study

Human: with this image, if you want to take off clothes and hang them in the closet, write your plan according to the image

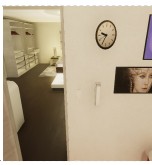

Assistant:
1. Find a location in the image where there is an empty hanger.
2. Identify the size or shape of the hanger to ensure it can accommodate your clothes.
3. Use the handheld device to remove the clothes from the image.
4. Position the clothes on the hanger in the image, ensuring they fit comfortably.
5. Once the clothes are positioned correctly, use the device to hang them on the hanger.
6. Once you have successfully hung the clothes, verify that they are visible in the image."

(a) Ask EmbodiedGPT to write the plan directly.

Human: with this image, if you want to take off clothes and hang them in the closet, identify where you need to go and what you need to do in the scene shown in this image.

Assistant:
1. To hang clothes in the closet, you need to identify the closet area or location within the room where you can place the clothes.
2. In the image, there is a closet visible on the far right side of the room. The location of the closet can vary depending on the specific setting or design of the room.
3. Once you have identified the closet, go to the closet and find a hanging rod or shelf to place the clothes on.
4. In the image, there is a hanging rod visible on the far right side of the door frame. Ensure that the hanging rod is large enough to accommodate the clothes you want to hang.
5. Take the clothes off the bed and place them on the hanging rod.

(b) Ask EmbodiedGPT to write the plan with ***image-related chain-of-thought***.

Figure 5: Performance Comparison with different types prompt.

[3] Tianhe Yu, Deirdre Quillen, Zhanpeng He, Ryan Julian, Karol Hausman, Chelsea Finn, and Sergey Levine. Meta-world: A benchmark and evaluation for multi-task and meta reinforcement learning. In *Conference on robot learning*, pages 1094–1100. PMLR, 2020.

[4] Suraj Nair, Aravind Rajeswaran, Vikash Kumar, Chelsea Finn, and Abhinav Gupta. R3m: A universal visual representation for robot manipulation. *arXiv preprint arXiv:2203.12601*, 2022.

[5] Rudolf Kruse, Sanaz Mostaghim, Christian Borgelt, Christian Braune, and Matthias Steinbrecher. Multilayer perceptrons. In *Computational Intelligence: A Methodological Introduction*, pages 53–124. Springer, 2022.

[6] Sergey Ioffe and Christian Szegedy. Batch normalization: Accelerating deep network training by reducing internal covariate shift. In *International conference on machine learning*, pages 448–456. pmlr, 2015.