# OpenReview forum: "EmbodiedGPT: Vision-Language Pre-Training via Embodied Chain of Thought"
_NeurIPS.cc/2023/Conference — NeurIPS 2023 spotlight_

### Official Review · Reviewer_eM5s · 2023-07-01

**Soundness:** 4 excellent
**Presentation:** 3 good
**Contribution:** 3 good
**Rating:** 8
**Confidence:** 4

**Summary:**

The given text discusses the field of embodied AI and its applications in robotics. It highlights the potential of large language models (LLMs) in enabling robots to process natural language instructions, perform multi-modal tasks, and plan actions in physical environments. However, the collection of high-quality embodied multi-modal data poses challenges due to the need for egocentric data and structured language instructions.

To address these challenges, the authors propose the creation of two datasets: EgoCOT and EgoVQA. EgoCOT is a large-scale embodied planning dataset that contains egocentric videos from the Ego4D dataset, along with carefully generated, filtered, and human-verified step-by-step language instructions. EgoVQA is an extension of the Ego4D dataset, focusing on egocentric human-object interaction video question answering tasks.
Based on these datasets, the authors present an end-to-end multi-modal embodied foundation model called EmbodiedGPT.

It consists of four integrated modules: a frozen vision model for encoding visual features, a frozen language model for natural language processing, an embodied-former for aligning visual and embodied instructions, and a policy network for low-level action generation. The model incorporates "chain-of-thought" planning capabilities and aims to enhance task performance and adaptability.

The contributions of this work include the development of the EmbodiedGPT model, the creation of the EgoCOT and EgoVQA datasets, and the demonstration of the model's effectiveness in various embodied tasks such as embodied control, embodied planning, video captioning, and video question-answering. EmbodiedGPT outperforms other models on benchmark tasks, showcasing its strong performance and advantages in generating reliable planning sequences and integrating high-level planning with low-level control.

**Strengths:**

Addressing a key challenge in embodied AI by creating new datasets: One major strength of the paper lies in recognizing a critical issue in embodied AI - the lack of good quality embodied multimodal data, especially those accompanied by structured language instructions. To tackle this issue, the authors created two new datasets - EgoCOT and EgoVQA which have the potential to significantly improve research in this area.

Proposing an effective solution with state-of-the-art results: Building upon these newly developed datasets, the authors introduced the EmbodiedGPT model. This model achieved impressive performance across multiple embodied tasks including embodied control, embodied planning, video captioning, and video question answering.

Focus on improving the interpretability of LLMs: Another notable aspect of the paper is its focus on improving the transparency and explainability of LLMs used in embodied AI. The authors designed the EmbodiedGPT architecture so that each module can be interpreted separately or jointly analyzed together with attention weights.

Encouraging open source collaboration through code release: Finally, the authors released their code publicly, fostering further research collaborations within the community.

**Weaknesses:**

Potential overfitting to specific domains: Despite being trained on diverse tasks, EmbodedGPT might have become too specialized in certain areas where the training datasets were biased towards, leading to suboptimal generalization abilities when facing unseen situations or domains.

Besides, the interpretability and explainability of the EmbodiedGPT may remain problematic, griven transformer-based training on language, vision, and action policy, which lacks explicit memory components to ensure accurate responses based on past experiences.

 Training EmbodedGPT requires substantial computational resources, making it less accessible to smaller organizations or individual researchers who cannot afford expensive hardware.

**Questions:**

What is the computational cost of training such a large model?

How it the long-term behavior of EMbodiedGPT?

How can embodiedGPT benefit real-world robotic applications?

**Limitations:**

 The authors adequately addressed the limitations

---

> ### Author Rebuttal · Authors · 2023-08-08
>
> We are immensely grateful for your invaluable efforts and insightful suggestions, which have been of great significance and played a crucial role in the improvement of our paper.
>
> **Q1.[Potential overfitting to specific domains]**
>
> Great question! Your insight highlights the very reason we constructed the EgoCOT dataset, following the same format as the general Visual Question Answering (VQA) task. By aligning the structure in this way, it allows for the joint training of both the EgoCOT data and other general vision-language datasets, such as MS-COCO captions. This approach ensures that EmbodiedGPT incorporates a more diverse range of data, enhancing its ability to generalize across various tasks.
>
>
> **Q2.[interpretability and explainability]**
>
> we recognize that it is an inherent challenge in dealing with large multi-modal models. We are actively investigating methods and chain-of-thoughts to enhance understanding, such as incorporating more explicit memory and reasoning, either from multi-frame/video or from symbolic language. This includes generating textual forms from symbolic graphs, which can provide more insights about the reason for decision-making, using language models and updating them continuously with new observations.
>
> There is an example:
>
> symbolic graph:
> ```python
> {
>   "Central node": {
>     "name": "Front view",
>     "features": {
>       "sink": "white",
>       "mirror": "above sink",
>       "atmosphere": "well-lit",
>       "trash can": "next to sink",
>       "floor": "blue tiled"
>     },
>     "connected_nodes": ["Left view", "Right view"]
>   },
>   "Left view": {
>     "name": "Left view",
>     "features": {
>       "urinals": "3 in a row",
>       "stalls": {
>         "left": 2,
>         "right": 1
>       },
>       "walls": "lined with white tiles",
>       "window": "above urinals",
>       "atmosphere": "spacious, well-lit"
>     },
>     "connected_nodes": ["Central node"]
>   },
>   "Right view": {
>     "name": "Right view",
>     "features": {
>       "sinks": "2 white sinks next to mirror",
>       "bottles": "2 (possibly containing hand sanitizer or other cleaning agents)",
>       "hand dryer": "mounted on the wall above the sinks"
>     },
>     "connected_nodes": ["Central node"]
>   }
> }
> ```
> Question: To find a flush toilet, you need to "move forward" "turn left" or "turn right", tell me your decision and tell me why?
>
> EmbodiedGPT: Based on the observations, it appears that the toilet stalls are likely to contain flush toilets. In the "left view" description, it was mentioned that there are stalls in the back of the bathroom. As such, to find a flush toilet, you should "move forward" and then "turn left".
>
> Sincerely thanks for your insightful suggestion again.  Improving transparency remains a vital direction that we are committed to pursuing in our future work.
>
> **Q3.[What is the computational cost of training such a large model?]**
>
> The computational cost of EmbodiedGPT for jointly training the model using the EgoCOT dataset, web video dataset, and COCO caption dataset was effectively managed on 8 A100 GPUs within a span of 16 hours. We have implemented a feasible and efficient method to train large models via a technique that combines a frozen module and prefix tuning. This approach significantly reduces the need for computational resources compared to traditional methods.
>
> **Q4.[What is the long-term behavior of EmbodiedGPT?]**
>
> Thank you for your insightful suggestion! We have added experiments on rearrangement tasks that require manipulation and interaction with multiple objects to evaluate the long-term performance of EmbodiedGPT. Specifically, the robot is tasked to organize scattered items on a table into a box using the Franka Emika Robot Arm. The performance and demo are demonstrated in **Figure 2 of rebuttal PDF** in the General Response section. Furthermore, as we show in Q2, EmbodiedGPT can also maintain a text-based symbolic graph, continuously updating this text graph based on the latest observations. This feature can further enhance its success rate in tasks that require long-sequence memory dependence, such as navigation tasks across multiple rooms.
>
> **Q5.[How can EmbodiedGPT benefit real-world robotic applications?]**
>
> Good question! EmbodiedGPT provides a strong foundation tool that robots can leverage to efficiently adapt to new instances of these tasks.  And can benefit real robot applications in the following points：
>
> 1. Improved Object Recognition and Interaction: The combined vision-language understanding of EmbodiedGPT allows robots to not just visually recognize objects, but also interpret their meaning and function based on language context. This enables more accurate object identification and more sensible interactions aligned with human intents.
> 2. Enhanced Human-Robot Communication: By processing both visual and natural language inputs, EmbodiedGPT can enable more intuitive communication between humans and robots. Robots can interpret complex commands involving speech and gestures and provide feedback in a more natural, human-like way.
> 3. Advanced Task Execution: EmbodiedGPT's joint understanding of visual scenes and language instructions empowers robots to perform tasks requiring both, such as "pick up the red apple on the left". This demands recognizing the apple visually while comprehending the language-based directions.
> 4. Adaptive Learning: Robots can leverage EmbodiedGPT to learn new skills by observing human demonstrations and explanations. Watching a person perform a task while listening to narrations allows adaptable learning to replicate the task.
>
> We are actively exploring these application directions and aim to demonstrate concrete real-world results in our future work. EmbodiedGPT provides a strong foundation for advancing robots' abilities to perform diverse skills in human-centric environments.
>
> \
> Thanks again for your time and effort! For any other questions, please feel free to let us know during the rebuttal window.

---

> > ### Comment · Reviewer_eM5s · 2023-08-10
> >
> > Thanks for the additional details provided. I will take them into consideration.

---

### Official Review · Reviewer_5cDX · 2023-07-02

**Soundness:** 2 fair
**Presentation:** 2 fair
**Contribution:** 2 fair
**Rating:** 3
**Confidence:** 5

**Summary:**

This paper proposes an end-to-end multi-modal foundation model for embodied AI. Specifically, the authors create a dataset termed EgoCOT from the Ego4D dataset and introduce a paradigm for both the vision and embodied tasks using large language models. However, I have some concerns about this paper. My detailed comments are as follows.

**Strengths:**

1、The paper well combines the large language models with embodied planning.
2、The proposed EgoCOT dataset and EgoVQA dataset are helpful for downstream tasks such as embodied planning and VQA.

**Weaknesses:**

To me, this paper is close to being interesting but I, unfortunately, do feel there are some weaknesses that, currently, prevent me from recommending acceptance.
1、The main idea has limited technical contribution:
The entire paper reads like a technical report and many parts of it are not coherent. The network architecture level in this regard can be said to have no innovative contribution. The frozen modules in the paradigm have been proposed and used in many papers related to large language models. Essentially, embodied former is also actually a cross-attention network used for visual and language modality interaction. As for the policy network, it simply encodes vision and combines it with multi-modal interaction output as input. I understand this paper has the purpose to use large language models for embodied tasks. However, at the method level, I see more about how to interact and align the image or video modality with the text modality, and in the experimental part, it also focuses on reflecting the ability of the method in image caption and object grounding, not even in some important experiment on embedded tasks (such as embedded navigation, embedded rearrangement, Instruction Following and Dialogue, etc.)

2、As a side note, I would like to let the authors know about some flaws in the presentation of the paper:
（1） In Section 3.1, in the second stage of the training, how are the language projection and prefix language adapter updating? I suggest that the authors explain this in detail.
（2） In Section 3.1, in the third stage of the training, does the Conv3D module need to be learned? And how to decide the time offset to be 2? I didn’t see any explanation or ablation studies of these hyperparameters.
（3） In Section 3.4, how to decide the start and end time points of a clip? Please explain the relationship of  \beta and \alpha and why the \beta/2*\alpha becomes the bound.
（4）I suggest adding more descriptions of the tasks conducted in the experiments. For example, I am confused about that why are the indicators Plan Reasonable and Plan Executable related to image caption tasks from MS-COCO.

**Questions:**

See the weaknesses.

**Limitations:**

See the weaknesses.

---

> ### Author Rebuttal · Authors · 2023-08-08
>
> Thanks for your efforts to improve the quality of our paper. We appreciate the flaws you identified in our paper's presentation and we believe they have been thoroughly clarified and addressed as follows.
>
> **Q1.[Limited novelty on architecture.]**
>
> We would like to clarify that our approach's novelty lies principally in the embodied chain of thought training methodology and the EgoCOT dataset, not the architectural design. We provide a  generic multimodal pretraining with the chain-of-thought strategy applicable to most existing model architectures. The core focus is exploring how chain-of-thought training can impart understanding of object interactability - for example, grasping a handle to open a door. Such knowledge further enables generating executable, high-quality plans. In manipulation tasks, while other multimodal models tend to concentrate on the detection of objects present on a table, our focus shifts toward a more nuanced understanding.
>
> Specifically, through the pre-training method guided by the embodied chain of thought, we endeavor to allow the multimodal system to comprehend how each object can be manipulated, as well as the intricate steps involved in a particular task. This distinct emphasis enables a more profound grasp of object interactions, fostering a comprehensive view of the task at hand, rather than simply identifying the presence of objects. It represents an advancement in modeling object manipulation and provides a more enriched understanding of complex task execution.
>
> As Review BD8Q commented, architecture is a minor point as training large models is a significant contribution in itself. We present these architectures not to emphasize their novelty, but rather their utility in making large language model pretraining viable for embodied goals with modest resources.
>
> **Q2.[Add more experiment on embodied tasks such as embodied navigation, embodied rearrangement, Instruction Following and Dialogue, etc.]**
>
> Thank you for your suggestion. We have added experiments on embodied navigation and embodied rearrangement. For navigation, we are focusing on the object navigation task of locating a flush toilet under visual inputs. See details in **Figure 1 of rebuttal PDF** at General response. Our experiments demonstrate that EmbodiedGPT offers robust priors for navigation tasks and can infer the most probable direction based on its knowledge. In the rearrangement task, we are using a Franka robotic arm to clean up cluttered items on a table and place them into a box.  See details in **Figure 2 of rebuttal PDF**.
> As for the Instruction Following and Dialogue tasks, our experiments in the original paper already utilize human language instructions as input to accomplish specific tasks, and they support multi-turn dialogue. See Fig. 4 in our main paper and Fig. 2 in the Appendix for examples.
>
> **Q3.[ In Section 3.1, in the second stage of the training, how are the language projection and prefix language adapter updating? I suggest that the authors explain this in detail. ]**
>
> In the second stage of training, the language projection layer and prefix language adapter are updated through backpropagation during pre-training. This allows the model to better capture complex language and reasoning skills.
>
> **Q4.[ In Section 3.1, in the third stage of the training, does the Conv3D module need to be learned? And how to decide the time offset to be 2?]**
>
> 1) Yes, the Conv3D module is learned during the pre-training process to extract relevant temporal information.
> 2) Regarding the choice of time offset being 2, it's mainly due to the constraints on computational resources and considerations for online inference speed. Our video input is limited to only 8 frames, and the time offset generally has two possible values: 2 and 4. We found that a time offset of 4 led to an inadequate understanding of the video content due to the larger span. Therefore, we selected 2 as the time offset. We acknowledge that with increased computational resources, the number of frames in the video input could be increased, which might lead to better performance, and there would be more options available for the time offset.
>
> **Q5.[In Section 3.4, how to decide the start and end time points of a clip? Please explain the relationship of $\beta$ and $\alpha$ and why the $\beta/2\*\alpha$ becomes the bound.]**
>
> Sorry for the confusion. $\beta_i$ is defined as the average temporal distance between consecutive narrations in a video. $\alpha$ is the average of all $\beta_i$ across the dataset. So $\beta_i/2\*\alpha$ determines the start and end time of a clip based on the timestamp $t_i$ of narration $T_i$. This creates a clip centered around the narration's timestamp with some padding. As shown in our paper, we use the same framework in the EgoVLP paper [A] to clip the video segment, and they have provided the ablation on the choice of clip strategy in their paper's Table 2.
>
> [A] Qinghong Lin, Kevin, et al. "Egocentric Video-Language Pretraining." NeurIPS 2022.
>
> **Q6.[Confused about that why are the indicators Plan Reasonable and Plan Executable related to image caption tasks from MS-COCO]**
>
> The utilization of the MS-COCO image captioning task in our experiments is aimed at testing the model's general visual language understanding ability and to provide comparisons with other models. It is not specifically designed to evaluate the indicators "Plan Reasonable" and "Plan Executable." These two indicators are not related to the image captioning tasks from MS-COCO, and we appreciate your feedback in highlighting this confusion. Further clarification will be added to the manuscript to ensure the distinction between the different tasks and evaluation metrics is clearer.
>
> \
> We wish that our response has addressed your concerns, and turns your assessment to the positive side. If you have any questions, please feel free to let us know during the rebuttal window. We appreciate your suggestions and comments! Thank you!

---

> > ### Author Response · Authors · 2023-08-13
> > **Sincerely looking forward to more discussion with you**
> >
> > Dear Reviewer 5cDX,
> >
> > We deeply appreciate the time you dedicated to reviewing our work and the insightful feedback you provided. We have crafted detailed responses and updated our results based on your comments, aiming to address all your concerns.
> >
> > We genuinely hope to engage in further dialogue to ensure that we have fully met your expectations. If any aspects of our work remain unclear, or if you have additional concerns, please don't hesitate to communicate them. We remain committed to providing clarity and addressing all feedback. We are always ready to address your further concerns and improve our paper.
> >
> > Best regards,
> >
> > The Authors.

---

> > ### Author Response · Authors · 2023-08-18
> > **Would you kindly consider raising the score?**
> >
> > Dear reviewer 5cDX,
> >
> > Thank you again for your insightful suggestions to improve this work. We appreciate you taking the time to provide detailed and constructive feedback.
> >
> > Regarding your concern about the limited novelty of the architecture, we wish to clarify that the core innovations of our approach lie in the embodied chain of thought training methodology and the EgoCOT dataset, rather than the model architecture itself. Based on your recommendation, we have added experiments on embodied navigation and rearrangement tasks. These help showcase our model's ability to leverage its knowledge to infer sensible directions and manipulate objects properly. As for dialogue and instruction following, our original experiments already incorporated human language inputs to accomplish goals across multiple turns. We have also provided more implementation details around the training process and hyperparameters based on your questions. We will include them in our final paper.
> >
> > As the rebuttal period is ending soon, we wonder if our response answers your questions and addresses your concerns. If yes, would you kindly consider raising the score? Thanks again for your very constructive and insightful feedback!
> >
> > Best, \
> > Authors

---

> ### Comment · Reviewer_5cDX · 2023-08-19
>
> Thanks for the responses of the authors. So far, the authors' reply has not satisfied me. Here are my reasons:
>
> 1.	As for the novelty of the paper, I still believe that this paper does not have enough contributions and novelty. The author claims that “…The core focus is exploring how chain-of-thought training can impart understanding of object interactability...while other multimodal models tend to concentrate on the detection of objects…”, which is extremely incorrect and has not been rigorously investigated. In fact, several previous works have explored how to use LLM to embodied tasks. Specifically, they also all use the prompt engineer and chain of thought, such as [1][2]. Moreover, please **DO NOT** try to avoid and refute the novelty of the architecture. Review BD8Q is also concerned about the novelty of the architecture. As far as the above is concerned, I really don't see the novelty and value of this paper, but it makes me feel that the author is overclaiming.
>
>
> 2.	Regarding the newly added embodied task experiments, the authors disappoint me even more. I have read the pdf of the rebuttal. I think these experiments are very tricky, making no comparison with the SOTA methods and it is difficult for me to see any in-depth analysis, especially how to reflect the role and novelty of the method proposed in this paper in the tasks. Take embedded navigation as an example, the benchmarks in this community are very mature now. The authors should conduct their method in the strong benchmarks (such as R2R, RXR, VLNCE, etc.) of VLN to compare and analyze with existing SOTA methods and give the quantified data, similar to NavGPT [2]. However, the authors gave a so-called "navigation" demo containing a trajectory composed of photographed pictures. As a reviewer, I don't even know what the setting is and whether the setting is reasonable, where the data comes from, and how the method of the paper works in it. There is also another reviewer (i.e., Reviewer P2YP) who questions the setting of this paper for the embodied tasks. I strongly recommend the authors make a comprehensive survey and understanding of the embodied community. And give a comparison of the **quantitative results** of the **SOTA methods** for each embodied task I mentioned. Otherwise, I think the authors cannot simply put the "embodied" hat on his method.
>
> Thanks again for your time for reading my questions.
>
> [1] Wu J, Antonova R, Kan A, et al. Tidybot: Personalized robot assistance with large language models[J]. arXiv preprint arXiv:2305.05658, 2023.
>
> [2] Zhou G, Hong Y, Wu Q. NavGPT: Explicit Reasoning in Vision-and-Language Navigation with Large Language Models[J]. arXiv preprint arXiv:2305.16986, 2023.

---

> > ### Author Response · Authors · 2023-08-20
> >
> > Dear reviewer 5cDX,
> >
> > Thank you for your response and suggestions. We respectfully push back your premise that novelty of the architecture should be the main consideration. We believe our multi-modal chain-of-thought training, end-to-end embodied control, and our EgoCoT dataset are strong contributions. Just as LLama2 [1] saw significant performance gains over LLama [2] by incorporating higher quality data and thereby greatly benefited the community, we believe our EgoCOT dataset offers a valuable contribution to the field of embodied AI.
> >
> > Moreover, the papers [3, 4] you mentioned were either published on arXiv after the NeurIPS submission timeline or overlapped with it. We believe these should not be used as grounds to diminish our contribution.
> >
> > We agree with you that more experiments on strong VLN benchmarks would be beneficial to our work, and we will surely include them as our future work. However, the focus of our work is pioneering research on how to build a general multimodal embodied AI foundation model that can serve as a basic tool for multiple tasks, rather than achieving SoTA performance on all benchmarks, which is out of our current scope. In this work, we referred to the most representative embodied large model work Palm-e [5] from Google, and adopted diverse manipulation tasks to validate the model's effectiveness. We also provided ample feasibility studies of applying EmbodiedGPT to real-world navigation, rearrangement and other tasks, to demonstrate its versatility.
> >
> > Thank you for your time and consideration.
> >
> > [1] Touvron H, Martin L, Stone K, et al. Llama 2: Open foundation and fine-tuned chat models[J]. arXiv preprint arXiv:2307.09288, 2023.
> >
> > [2] Touvron H, Lavril T, Izacard G, et al. Llama: Open and efficient foundation language models[J]. arXiv preprint arXiv:2302.13971, 2023.
> >
> > [3] Wu J, Antonova R, Kan A, et al. Tidybot: Personalized robot assistance with large language models[J]. arXiv preprint arXiv:2305.05658, 2023.
> >
> > [4] Zhou G, Hong Y, Wu Q. NavGPT: Explicit Reasoning in Vision-and-Language Navigation with Large Language Models[J]. arXiv preprint arXiv:2305.16986, 2023.
> >
> > [5] Driess D, Xia F, Sajjadi M S M, et al. Palm-e: An embodied multi-modal language model[J]. arXiv preprint arXiv:2303.03378, 2023.
> >
> > Sincerely, \
> > Authors

---

### Official Review · Reviewer_P2YP · 2023-07-05

**Soundness:** 4 excellent
**Presentation:** 4 excellent
**Contribution:** 4 excellent
**Rating:** 8
**Confidence:** 5

**Summary:**

In this work, the authors propose an end-to-end multi-modal model for Embodied AI field. Based on the Ego4D dataset, the authors build a large-scale Embodied planning dataset, i.e., EgoCOT, which contains machine-generated language instruction with chain-of-thoughts mode. Besides, an EmbodiedGPT based on LLM is proposed to achieve the plan generation task, where the corresponding training approach is designed for prefix tuning on EgoCOT. Experimental results demonstrate that EmbodiedGPT can achieve the promising ability of plan generation.

**Strengths:**

- This work provides the EgoCOT dataset with EgoVQA as an extension, which can promote the development of LLM in the Embodied AI field.

- This work makes the pioneering attempt to build a foundation model with the chain-of-thought capability. Building the framework based on the Vision Transformer and LLaMA, and designing the Embodied-Former as the bridge between the visual and language modality is reasonable.

- Extensive quantitative and qualitative results are provided for examining the model's effectiveness. The results show that EmbodiedGPT can produce reasonable answers to describe what happened in the input video and generate the step-by-step action sequence.

- The paper is well written and easy to follow with detailed instructions about how to construct and train the model.

**Weaknesses:**

- In the field of Embodied AI, an important task is \*agent navigation*, but this work does not cover this part. I understand that the authors focus on the multi-modal foundation model, and there is a vision-language navigation (VLN) task [1][2] that requires the agent to conduct navigation according to the given language instruction. Thus I believe the VLN is a promising task for authors to test the model's navigation ability, e.g., verifying if the model can generate step-by-step navigation instructions with chain-of-thoughts.

- I am confused about the setting of EmbodiedVQA in this paper. In my view, EmbodiedVQA [3] is actually the task that embodied agents navigate in rich 3D environments in order to answer the given question.

- Currently, the model can generate detailed descriptions of a given video and convert them into planning steps with chain-of-thoughts. However, can the model generate executable steps for completing a specific task based solely on a static image of an object, rather than a demonstration video?

- The authors should provide more details on how to use the model for Embodied control, specifically how to convert the text output into step-by-step control signals, as this is a crucial aspect for implementing embodied agents.

- Minor: the colors of Figure 2 are a bit chaotic, and if there is no special meaning, I suggest it can be simplified.

[1] Vision-and-Language Navigation: Interpreting visually-grounded navigation instructions in real environments. In CVPR 2018.

[2] Reverie: Remote embodied visual referring expression in real indoor environments. In CVPR 2020.

[3] Embodied Question Answering. In CVPR 2018.

**Questions:**

See Weaknesses.

**Limitations:**

I do not foresee obvious undesirable ethical or social impacts at this moment.

---

> ### Author Rebuttal · Authors · 2023-08-08
>
> We sincerely appreciate the time and effort you have invested in reviewing our paper. Your meticulous evaluation and insightful recommendations have significantly contributed to improving the quality of our research.
>
> **Q1.[Evaluating on vision-language navigation (VLN) tasks to test the navigation ability.]**
>
> Thank you for your insightful suggestion regarding the exploration of vision-language navigation (VLN) tasks with EmbodiedGPT. We concur that VLN is a fascinating direction for future investigation, particularly as it involves the generation of navigation instructions.
>
> To that end, we have conducted an experiment on the vision-language navigation task using EmbodiedGPT. The task chosen was an object-finding exercise, guided by the human instruction, "please find a flush toilet." EmbodiedGPT processed both historical and current observations to infer the direction needed to locate the target object. The performance details can be found in **Figure 1 of rebuttal PDF**.
>
> EmbodiedGPT demonstrates the ability to generate step-by-step navigation instructions, employing a coherent chain of thought when provided with language instructions and visual observations of new environments. We appreciate this recommendation and plan to extend EmbodiedGPT's navigation capabilities, benchmarking it on VLN tasks in future research.
>
> **Q2.[Confused about our embodied VQA setting compared to prior work.]**
>
> Sorry for the confusion. We would like to clarify that the Embodied VQA tasks presented in our paper are specifically focused on egocentric vision question answering for both embodied tasks and general recognition tasks, such as video/image recognition and task-oriented planning. This approach distinguishes itself from navigation-based EmbodiedVQA tasks as defined in prior work, which necessitates navigating complex 3D environments. We will revise the paper to make our setting clearer. Once again, we thank you for pointing out this crucial aspect.
>
>
> **Q3.[If the model can generate plans from static images without demonstration videos.]**
>
> As illustrated in the appendix figures (Figure 3 and Figure 4), EmbodiedGPT can generate plans based solely on static images of environments or objects, without the necessity of demonstration videos. While videos may offer supplementary guidance, the model is fully capable of formulating reasonable plans utilizing static images alone. To further substantiate this capability, we have included additional experiments on an embodied rearrangement task, as demonstrated in **Figure 2 of rebuttal pdf**, EmbodiedGPT can also generate clear plans with static images.
>
> **Q4.[More implementation details on converting text plans to control signals.]**
>
> Specifically, we leverage cross-attention modules in the Embodied-Former to extract features relevant to the plan from the encoded visual observations. These plan-relevant features are then passed to the policy network, which maps them to parameterized low-level actions and control signals, such as gripper poses and joint velocities, that can be executed to perform the task.
>
>
> \
> Thanks again for your time and effort! For any other questions, please feel free to let us know during the rebuttal window.

---

### Official Review · Reviewer_BD8Q · 2023-07-07

**Soundness:** 3 good
**Presentation:** 2 fair
**Contribution:** 3 good
**Rating:** 7
**Confidence:** 4

**Summary:**

This paper introduces EmbodiedGPT, a new multi-modal foundation model designed for application to embodied (robotic) tasks. EmbodiedGPT combines existing large pretrained (and frozen) vision and language models (namely the ViT-G/14 EVA-CLIP vision model and LLaMA-7B) through the use of an "Embodied-Former" which converts input text tokens and vision features into prefix tokens that can be fed directly to the LLaMA model. To pretrain this Embodied-Former model, this work introduces two new datasets (EgoCOT and EgoVQA), which contain video clips taken from the Ego4D dataset annotated with text captions generated by ChatGPT that, for EgoCOT, describe chain-of-thought-style plans corresponding to the actions taken within the clip, and, for EgoVQ, include questions and answers related to the clip. Using these new datasets and a 3-stage training pipeline, this paper trains the EmbodiedGPT model and, via human evaluations, show that humans generally prefer the captions and embodied plans generated by EmbodiedGPT even when compared to a LLaVA-13B baseline which has many more parameters. Finally, the paper shows that, by attaching an MLP policy head to the output of the Embodied-Former, it is possible to use imitation learning from a small set of demonstrations to train performant physical-manipulation agents in simulation. Experiments show that EmbodiedGPT outperforms ablations of itself and a R3M-based SoTA model for these tasks.

**Strengths:**

This paper studies a very popular, important, and challenging area of work: building foundational models for robotic control and embodied world understanding. The proposed EmbodiedGPT model represents a modest but non-trivial step towards this grand goal. In my view, the major contributions of this are the substantial efforts taken to create the EgoCOT/EgoVQA datasets and execute the complex and computationally expensive 3-stage strategy required for training the EmbodiedGPT model. Generating these datasets and training the model clearly required significant effort and the experimental results appear quite strong.

**Weaknesses:**

While I lean positive for this paper, I will highlight some weaknesses below that temper my excitement.

## Statistical significance of the human results

The only evaluation of EmbodiedGPT against other existing large vision-and-language models comes in a 30-participant human study using 20 tasks (10 captioning, 5 simulated planning, 5 real planning). As this sample size is quite small, and the performance difference sometimes relatively minor, it's challenging to draw meaningful conclusions from the results in Table 1. Moreover, it is hard to know how representative the 20 tasks are of general captioning/planning tasks, how were these chosen?

There are ways to provide a statistical analysis of this data but they are not trivial: you'd need to account for the correlated nature of the data (e.g. that individual participants rated multiple examples and that performance may be correlated with the task type and within each question). While not a perfect treatment, at the very least a paired t-test could be used to compare EmbodiedGPT with the other models.

## Really "foundational"?

While EmbodiedGPT is touted as a "foundation model for embodied AI", the results do not quite support this in my view. In particular, I feel that the EmbodiedGPT misses the critical capacity to handle partial observability and I was surprised not to see any real-world examples of robotic control, more details below.

### Lack of tasks involving partial observability

The only embodied tasks evaluated in this paper involve static robotic arms acting in fully observed (or almost fully observed) environments. There are many embodied tasks (e.g. visual navigation) for which partial observability plays a critical role. From my understanding, the policy component of the EmbodiedGPT model makes decisions from a single visual observation in isolation from all previously seen states and thus would be expected to fail in all such tasks.

### Lack of application to real-world embodied tasks

While it is great to see that EmbodiedGPT outperforms the R3M model in the simulated environments, one of the strengths of the R3M paper was demonstrating that it also functioned well when applied to real robots. While not necessary for acceptance, it's difficult to call EmbodiedGPT a robotic foundation model without at least some real-world results.

## Architectural novelty

This is a very minor point as I consider the training of large models to be a significant contribution in itself, but the proposed EmbodiedGPT architecture is not especially novel: it combines popular pretrained vision/language models with well-known prefix tuning ideas. While you cite Tsimpoukelli et al. (2021), I was a bit surprised to see more discussion of how their work relates to the proposed approach. Moreover, simply searching for "embodied" in the citations of the above paper (i.e. https://scholar.google.com/scholar?hl=en&as_sdt=5%2C48&sciodt=0%2C48&cites=16154696122208258147&scipsc=1&q=embodied) appears to surface several additional citations that should be included in this work.

**Questions:**

I have listed some of my major questions and weaknesses above. If some of these weaknesses can be addressed (especially the point about statistical significance) I expect to increase my score. In the below I include, more minor, line-by-line comments and questions.

Line 60
- "EmbodiedGPT ’s" should be "EmbodiedGPT’s". I suspect this is a problem due to latex macros, try adding {} after your macro inline, i.e. \mymodel{}.

Line 77
- This "200M annotated videos" statement seems like it must be wrong. Later on, you note that there are 2927 hours of video in total, this is approximately 10M seconds of video which would correspond to ~200M frames depending on the framerate, is that what you mean?

Figure 2
- This figure could use some work, for instance, it's very unclear how videos are being input into the model.

Lines 146-147
- What is the value of the "learnable embodied queries"? What makes them "embodied"?

Lines 172-174
- I found this description of the video encoding procedure to be too vague, please add more details.

Lines 181-204
- This entire discussion could be simplified and better presented, the idea here is quite simple and much of the formalism is extraneous. For instance, you introduce a "mapping function (...) M: z -> z'" but this mapping function is just a linear projection and the "M" notation seems to be never used again. You could just say "z is then mapped to z' by via a linear projection."

Line 241
- Extra space after the comma in 9,645.

Lines 225-228
- How is \beta_i "adjustable" if it's defined in this way? It seems to be a fixed function of other fixed values.
- Why not just set [t^{start}_{i}, t^{end}_{i}] to be [(t_{i-1}+t_{i})/2, (t_{i+1} + t_{i})/2]?

Figure 3
- This seems like a surprising qualitative example to choose as the EmbodiedGPT model says many wrong things. E.g. (1) the clock is not actually above the door, (2) the coat hanger is not on the floor, and (3) the coat hangers can be seen towards the top left, not the "near the right side of the doorway".

Figure 4
- Again, this is a surprising example as the video shows it lowering the ring over the cylinder twice whereas EmbodiedGPT seems to only suggest this happens once.

**Limitations:**

I would have liked to see some discussion of the limitations I discussed in the _Really "foundational"?_ section of the weakness. Otherwise, limitations are reasonably well addressed.

---

> ### Author Rebuttal · Authors · 2023-08-08
>
> We are deeply grateful for your careful review and meticulous line-by-line suggestions!
>
> **Q1. [It is hard to know how representative the 20 tasks are of general captioning/planning tasks, how were these chosen?]**
>
> To ensure comprehensive representation, the tasks are selected across real-world and simulated settings (Virtual Home, Habitat, and AI2THOR) that span various indoor areas like kitchens, bedrooms, and offices. These tasks address key interaction and cognitive elements, from basic actions like "how to wash a vegetable" to complex reasoning questions indicative of decision-making and planning.
>
> **Q2. [User study's sample size and lack of statistical significance testing in human study.]**
>
> We've conducted a Welch t-test comparing EmbodiedGPT to the baselines. The table below presents the T-statistic and P-value derived from the responses of 30 individuals. A p-value below 0.05 signifies a statistically significant difference between the groups. EmbodiedGPT demonstrates a statistically significant improvement over the baselines, underscoring its effectiveness.
> | Scoring Standard | Groups Compared | T-statistic | P-value  | Significant |
> |------------------|-----------------|-------------|----------|-------------|
> | Plan Reasonable  | Minigp4 vs EmbodiedGPT | -23.39 | 6.26e-30 | Yes |
> | Plan Reasonable  | LLaVA-7B vs EmbodiedGPT | -8.15  | 1.73e-11 | Yes |
> | Plan Reasonable  | LLaVA-13B vs EmbodiedGPT | -2.35  | 0.022    | Yes |
> | Plan Executable  | Minigp4 vs EmbodiedGPT | -24.74 | 3.90e-31 | Yes |
> | Plan Executable  | LLaVA-7B vs EmbodiedGPT | -9.19  | 2.61e-13 | Yes |
> | Plan Executable  | LLaVA-13B vs EmbodiedGPT | -2.12  | 0.038    | Yes |
>
> **Q3. [Lack of evaluation on tasks requiring partial observability.]**
>
> Good Question! We address your concerns in the following aspects.
> - **EmbodiedGPT's Design with Respect to Partial Observability**: EmbodiedGPT has been carefully designed to consider the problem of partial observability. It supports multi-frame input and is capable of understanding the changes and motions across different frames thereby handling occlusion. Specifically, through the use of EgoCOT training, it can easily capture the variations and partial observations in object states/motions.
> - **Additional Experiments**: We add experiments on the **visual navigation task** which aims to find the flush toilet with partial observation. The results show that EmbodiedGPT provides robust priors and efficiently consolidates historical observations into textual records for future decisions. See detailed results **Figure 1 of rebuttal PDF**  in General Response.
>
> **Q4. [Lack of application to real-world embodied tasks.]**
>
> We add a real-world experiment on the Franka Emika robot arm, focusing on a rearrangement task to pack bottles cluttering the table into a box. See the demo in the **Figure 2 of rebuttal PDF**.
>
> **Q5. [Architectural novelty.]**
>
> Thank you for recognizing that our training large models is a significant contribution in itself. We will conduct a broader literature review.
>
> 1. We'd like to clarify that the primary novelty of our work does not reside in the architecture design itself, but rather in the unique approach we have taken in the "embodied chain of thought" training paradigm and the data handling. The core emphasis of our work is on guiding the model to understand the interactivity of objects through intuitive knowledge, such as "if a door has a handle, one must grasp the handle to open it." This stands in contrast to conventional multimodal pre-trained models that mainly focus on grounding in common semantics.
> 2. The contributions we have highlighted concerning architecture design are not meant to stress the key novelty but rather to offer an efficient and feasible training method within these constraints.
>
> **Q6. [Video Input Mechanism into the Model]**
>
> Videos are processed by:
> 1. Keyframe Extraction: Extracting eight uniformly spaced keyframes.
> 2. Segmentation to 3D patches: Keyframes are fed into the Conv3D module, segmented into 3D patches each defined by `patch_size*patch_size*frame_stride`.
> 3. Patch Representation: These patches act as spatio-temporal cubes in the video, capturing both visual content and temporal sequence.
> 4. Token Conversion: Patches are transformed into visual tokens via Conv3D  with " kernel_size=(frame_stride, patch_size, patch_size), stride=(frame_stride, patch_size, patch_size) "and then fed into the vision transformer.
>
> **Q7.[What is the value of the "learnable embodied queries"? What makes them "embodied"?]**
>
> The learnable embodied queries are vector embeddings that interact with the visual features through cross-attention layers in the Embodied-Former module (Section 3.2). They are considered "embodied" because their goal is to extract only the most relevant visual information for embodied tasks like planning and control. These queries offer a compact representation with task-specific semantics and also reduce the input tokens to the language model(an image is encoded into far more visual tokens than text tokens).
>
> **Q8.[How is $\beta_i$ "adjustable".]**
>
> Sorry for the confusion. We would like to clarify that the discrepancies in the length of video segments corresponding to a single caption are inherent to the dataset itself. The calculation method we used is fixed, and we apologize for any confusion this may have caused.
>
> **Q9.[Minor issues and typos.]**
>
> We will revise our paper carefully according to your suggestions.
> As for Figure 3, spatial misalignment is prevalent in image captioning, even in state-of-the-art models. Scaling the model can enhance detail alignment and capacity. As for Figure 4, we apologize for the mistake in the order of the plotted frames. We appreciate your keen eye and patience, and we have revised it with the correct order in **Figure 3 of rebuttal PDF**.
>
> \
> Thanks again for your time and effort! For any other questions, please feel free to let us know during the rebuttal window.

---

> > ### Author Response · Authors · 2023-08-13
> > **Thanks for your great efforts and look forward to your reply**
> >
> > Dear reviewer BD8Q,
> >
> > Firstly, we'd like to express our profound gratitude for your previous suggestions. Your recommendations were meticulous and constructive, prompting deeper reflections on many issues. We truly value the effort and thought you've invested in our work.
> >
> > We deeply appreciate the time you dedicated to reviewing our work and the insightful feedback you provided. Following your comments, we've crafted detailed responses and made necessary updates, aiming to address all your concerns.
> >
> > We sincerely hope to further discuss with you and ensure that we've completely met your expectations. If any aspects of our work remain unclear, or if further concerns arise, please don't hesitate to share them. Our commitment remains steadfast in providing clarity and responding to all feedback.
> >
> > Best regards,
> >
> > The Authors.

---

> > > ### Comment · Reviewer_BD8Q · 2023-08-15
> > >
> > > Dear Authors,
> > >
> > > I very much appreciate the substantial effort you have undertaken in this rebuttal period: the addition of a real-world experiment is especially impressive! While I do believe there are some ways in which the paper could be further improved (e.g. the real-world and partially-observed experiments are still much more "qualitative" than they are "quantitative") I am generally satisfied with the responses and plan to increase my score.
> > >
> > > Best,
> > > BD8Q

---

> > > > ### Author Response · Authors · 2023-08-16
> > > > **Sincerely thank you for your support and recognition of our paper**
> > > >
> > > >
> > > > Dear Reviewer BD8Q,
> > > >
> > > > We sincerely thank you for your constructive suggestions and effort. We will add more quantitative results to our final version according to your recommendations. Thank you for your kind suggestions and warm help. Please don’t hesitate to let us know if there are any additional clarifications we can offer, looking forward to your post-rebuttal rating!
> > > >
> > > > Best,
> > > >
> > > > The Authors.

---

### Author Rebuttal · Authors · 2023-08-08

# General Response
We sincerely appreciate all reviewers’ time and efforts in reviewing our paper. We are glad to find that reviewers generally recognized our contributions:
* **Model Development.** Taking a pioneering step in developing a foundational model for embodied AI [BD8Q, P2YP], integrating large multi-modal models with embodied planning [5cDX].
* **Dataset Creation.** Introducing the EgoCOT dataset, along with the EgoVQA extension, to enhance the multi-modal chain-of-thought capabilities in the Embodied AI domain [BD8Q, P2YP, 5cDX]. This addresses a significant challenge in embodied AI through the generation of new datasets [eM5s].
* **Experiments.** Showing remarkable and promising results, demonstrating the effectiveness of generating high-quality step-by-step action sequences [BD8Q,P2YP], and achieving impressive performance across multiple embodied tasks [eM5s].
* **Writing.** Well-written and easy to follow [P2YP].

And we also thank all reviewers for their insightful and constructive suggestions, which help a lot in further improving our paper. In addition to the pointwise responses below, we summarize supporting experiments added in the rebuttal according to reviewers’ suggestions.

**New experiments**

- Additional **Visual Navigation** Experiment [BD8Q,5cDX]. We conducted an experiment to test EmbodiedGPT's capabilities in a **partially observed** setting with a visual object navigation task. Specifically, the task required the model to locate a target object (a flush toilet) using only limited visual inputs and a language command. Our results indicate that EmbodiedGPT provides valuable priors for visual navigation tasks. Even in instances where the observed images don't contain the target object, the model can deduce the most likely direction based on its knowledge. This suggests that, when paired with cutting-edge navigation algorithms, EmbodiedGPT could emerge as a formidable tool in this domain. Please see more details in Figure 1 of the following rebuttal PDF.

- **Real robot** experiment on **embodied rearrangement** task with Franka Emika robot arm [BD8Q,5cDX].  The video is provided in a separate link using "official comments" according to the strict regulation.  In the embodied rearrangement task, we are using a Franka robotic arm to clean up cluttered items on a table and place them into a box. Please see more details in Figure 2 of the following rebuttal PDF.

**New Data Analysis**
- Statistical significance testing of the human evaluation results, confirming the superior performance of EmbodiedGPT over baselines [BD8Q].

We believe these new additions help address reviewer concerns around partial observability, real-world evaluation, novelty, and presentation issues. We thank the reviewers' time and feedback in improving the quality of our work and we hope the revisions further highlight the contributions made. Please let us know if any clarification or additional experiments would further strengthen the paper. We would be happy to incorporate all these suggestions in the final version.

---

### Author Response · Authors · 2023-08-08
**Real robot demo of EmbodiedGPT**

Dear Area Chair and reviewers,

Thanks for your efforts in handling our manuscript and we would like to express our gratitude for the constructive suggestions provided by the reviewers.

In response to the request from reviewer [BD8Q,5cDX], we have conducted an experiment on a real robot, focusing on a typical embodied rearrangement task. The experiment was performed using a Franka Emika robot arm. Please find a demonstration video at the following link: https://www.dropbox.com/scl/fi/qrc2514rgwoalzg0419gc/rearrangement_demo_EmbodiedGPT.mp4?rlkey=1d2wea2g26tccy478vvwsnu5z&dl=0.

Best, \
The authors.

---

### Decision · Program_Chairs · 2023-09-21

**Decision:**

Accept (spotlight)

**Comment:**

The authors introduce both an approach and data for converting a video (e.g. Ego4D) to a task plan for robot execution.  This CoT approach helps the model build a representation of low-level skill sequences that improves performance.  The same underlying model can then also be used for VQA and related multimodal tasks.  The authors expanded their set of experiments in the rebuttal period -- all of which once incorporated will make the paper much stronger.  The reviewers are generally in agreement that the paper is in fact a useful and interesting contribution, there are a few rhetorical points (e.g. generally by BD8Q or where terms might be confused -- P2YP) that the authors should consider to strengthen the paper, reduce possible confusion, and scope the contributions.